# Strong yet flexible ceramic aerogel

Lei Su [1], Shuhai Jia [2] ✉, Junqiang Ren[3], Xuefeng Lu[3], Sheng-Wu Guo[1], Pengfei Guo[1], Zhixin Cai[1], De Lu[1], Min Niu[1], Lei Zhuang [1], Kang Peng [1] & Hongjie Wang [1] ✉

Ceramic aerogels are highly efficient, lightweight, and chemically stable thermal insulation materials but their application is hindered by their brittleness and low strength. Flexible nanostructure-assembled compressible aerogels have been developed to overcome the brittleness but they still show low strength, leading to insufficient load-bearing capacity. Here we designed and fabricated a laminated SiC-SiO$_x$ nanowire aerogel that exhibits reversible compressibility, recoverable buckling deformation, ductile tensile deformation, and simultaneous high strength of up to an order of magnitude larger than other ceramic aerogels. The aerogel also shows good thermal stability ranging from −196 °C in liquid nitrogen to above 1200 °C in butane blow torch, and good thermal insulation performance with a thermal conductivity of $39.3 \pm 0.4$ mW m$^{-1}$ K$^{-1}$. These integrated properties make the aerogel a promising candidate for mechanically robust and highly efficient flexible thermal insulation materials.

Developing thermal insulators that are simultaneously strong and flexible under compressive, tensile, and bending deformations is of great significance for the heat preservation and thermal protection of abysmal sea and aerospace vehicles. Ceramic aerogels are attractive thermal insulation materials, owing to their low density, low thermal conductivity, and good thermal stability[1–7]. However, conventional ceramic aerogels composed of oxide nanoparticles usually exhibit low strength and brittleness due to the weak intergranular necking junctions and brittle nature of ceramics[2,3]. To strengthen and toughen ceramic aerogels, polymer crosslinked ceramic aerogels[8,9] and fiber-reinforced ceramic aerogels[10,11] have been fabricated to achieve improved strength and deformability, however, thermal instability or severe dust release remain in these materials and impede their practical applications.

The recent development of flexible ceramic nanostructure-based aerogels provides another approach to overcome these drawbacks[12–18]. The good flexibility of ceramic nanostructure building blocks and the nanostructures-assembled highly porous microstructures enable these ceramic aerogels reversible compressibility, or even recoverable bendability[16] and stretch[17]. Three-dimensional printing has also been developed to fabricate resilient ceramic aerogel with tunable mechanical properties[19]. However, the strength and modulus of these modified aerogels are usually several to tens of kPa[6,7,12–19], which are too weak to provide adequate load-bearing capacity during the handling process and services. For example, thermal sealing materials for the hatch of a hypersonic aircraft not only require good thermal insulation performance and elasticity but also sufficient load-bearing capacity to withstand the external aerodynamics, vibrations, and the force during opening and closing of the hatch[20,21]. Increasing the density of ceramic aerogels is a facile way to improve their modulus and strength[22–24]. However, this strategy always results in a decrease in deformability[23,24] and thus an increase in security risk during services. Moreover, with the increase of density, the thermal conductivity of aerogels is usually increased significantly because of the increased contribution from solid conduction[14,23,25]. Therefore, developing ceramic aerogels with simultaneous mechanical robustness, adequate deformability, and good thermal insulation is highly desired but remains in challenges of resolving the conflicts among these properties.

In nature, many attractive materials such as silkworm cocoons are simultaneously lightweight, strong, tough, and thermally insulating[26–28]. Such attractive comprehensive properties of silkworm cocoon are attributed to its naturally evolved high-strength yet flexible silk[26,29] and the silk-assembled laminated microstructure[26–28]. On one hand, the aligned silk in each layer provides the cocoon with high

[1]State Key Laboratory for Mechanical Behavior of Materials, Xi'an Jiaotong University, Xi'an 710049, China. [2]School of Mechanical Engineering, Xi'an Jiaotong University, Xi'an 710049, China. [3]State Key Laboratory of Advanced Processing and Recycling of Non-ferrous Metal, Department of Materials Science and Engineering, Lanzhou University of Technology, Lanzhou 730050, China. ✉e-mail: shjia@xjtu.edu.cn; hjwang@xjtu.edu.cn

Young's modulus, strength, and toughness to protect the pupas from external attack[26]. On the other hand, the anisotropic thermal conducting behavior in the laminated microstructure helps it reduce heat flux in the direction perpendicular to the silk layers and thus gives it low thermal conductivity to provide a still internal temperature environment[27,28]. These characteristics make the silkworm cocoon one ideal model to overcome the above-mentioned conflicts, thus realizing the rational design of strong yet flexible ceramic aerogels with simultaneous good thermal insulation.

Here we report a laminated SiC-SiO$_x$ nanowire aerogel that exhibits mechanical robustness and flexibility under compressive, tensile, and bending deformations, good thermal insulation, and thermal stability in a wide temperature range. The laminated aerogel exhibits reversible compressibility, ductile tensile deformation, recoverable bendability, and recoverable buckling deformation. It shows a high compressive modulus of $222 \pm 32.7$ kPa, high compressive stress of $1255 \pm 116.3$ kPa at 80% strain, high tensile stress of $399 \pm 83.4$ kPa and modulus of $4855 \pm 111.0$ kPa, high bending strength of $261 \pm 11.4$ kPa, which are several to tens of times higher than other resilient ceramic aerogels[12–19,30], showing improved load-bearing capacity under diverse deformations. The flexibility also maintains under 1200 °C butane blow torch and at temperatures as low as −196 °C in liquid nitrogen. The thermal conductivity of the laminated aerogel is $39.3 \pm 0.4$ mW m$^{-1}$ K$^{-1}$, comparable to that of the silk cocoon[27,28], showing good thermal insulation. Such rarely reported integrated properties make the aerogel one of the promising candidates for highly efficient and mechanically robust thermal insulation materials.

## Results and discussion
### Preparation and characterization

We used a homemade highly compressible and stretchable SiC-SiO$_x$ nanowire aerogel paper[30] with a density of about 5.7 mg cm$^{-3}$ as the raw materials to prepare the laminated SiC-SiO$_x$ nanowire aerogel. The raw SiC-SiO$_x$ nanowire aerogel paper was prepared through a chemical vapor deposition method, which is illustrated in detail in Methods. To realize the construction of the laminated microstructure, a facile capillary force-induced self-assembly method was developed. Figure 1a illustrates the fabrication process of the laminated ceramic aerogel. Firstly, the raw aerogel paper was cut into pieces and then they were stacked layer-by-layer to form a bulk aerogel. The bulk aerogel was immersed in ethanol. After it was fully infiltrated, the aerogel was put out and then dried naturally. During the evaporation of ethanol in the drying process, the self-assembly of the nanowires induced by the capillary force took place, resulting in the formation of the laminated SiC-SiO$_x$ nanowire aerogel. Figure 1b shows the macroscopic morphology of the as-prepared laminated aerogel standing on the surface of a leaf, in which we can observe the laminated structure of the aerogel obviously.

To insights into the formation mechanism of the laminated structure, we attached a piece of the raw SiC-SiO$_x$ nanowire aerogel on a solid substrate (glass slide) mounted on a heating element with a temperature of 50 °C (Fig. 1c, d). Then we dropped ethanol on the aerogel paper and observed the microstructure evolution from the cross-section during the evaporation of ethanol. As shown in Fig. 1e and Supplementary Movie 1, during the drying process, a piece of aerogel scrap flows along the radial direction. This phenomenon

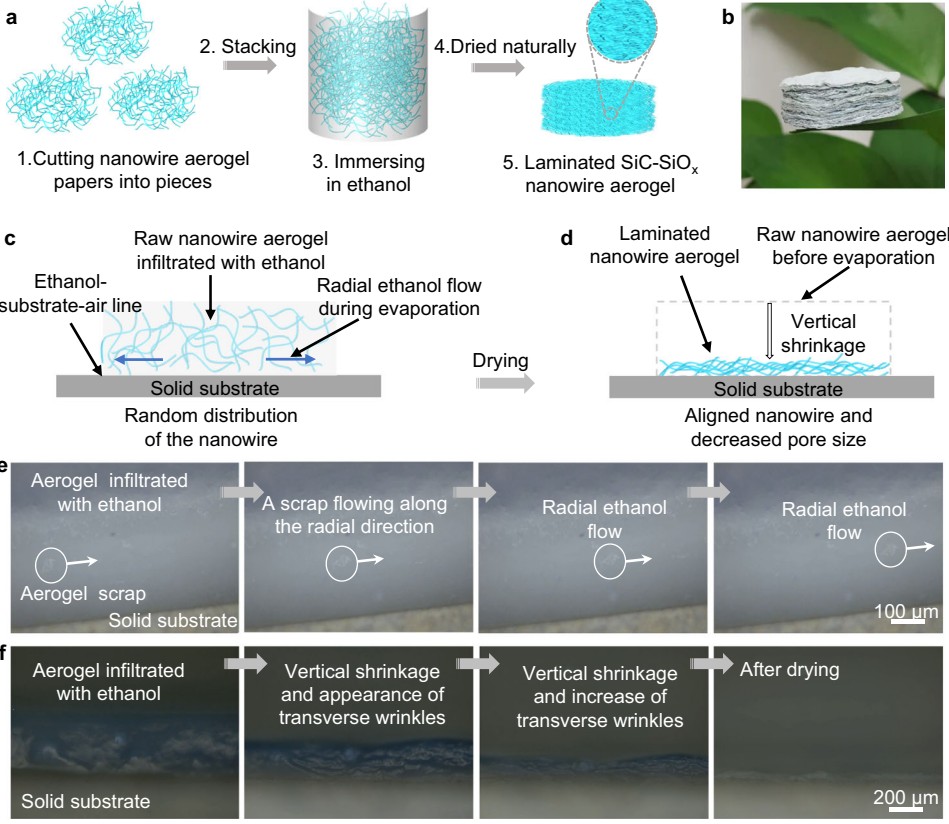

**Fig. 1 | Preparation process, formation mechanism, and macroscopic morphology of the laminated SiC-SiO$_x$ nanowire aerogel. a** Schematic illustration of the preparation process. **b** Macroscopic morphology of the aerogel. **c, d** Schematic illustration of the formation mechanism of the laminated aerogel during the drying process. The radial capillary flow and vertical shrinkage of the aerogel work together to result in the formation of the laminated aerogel. **e** Observation of the radial ethanol flow during the evaporation by using an optical microscope. **f** Optical microscopy images showing the vertical volume shrinkage and reorientation of the nanowires during evaporation.

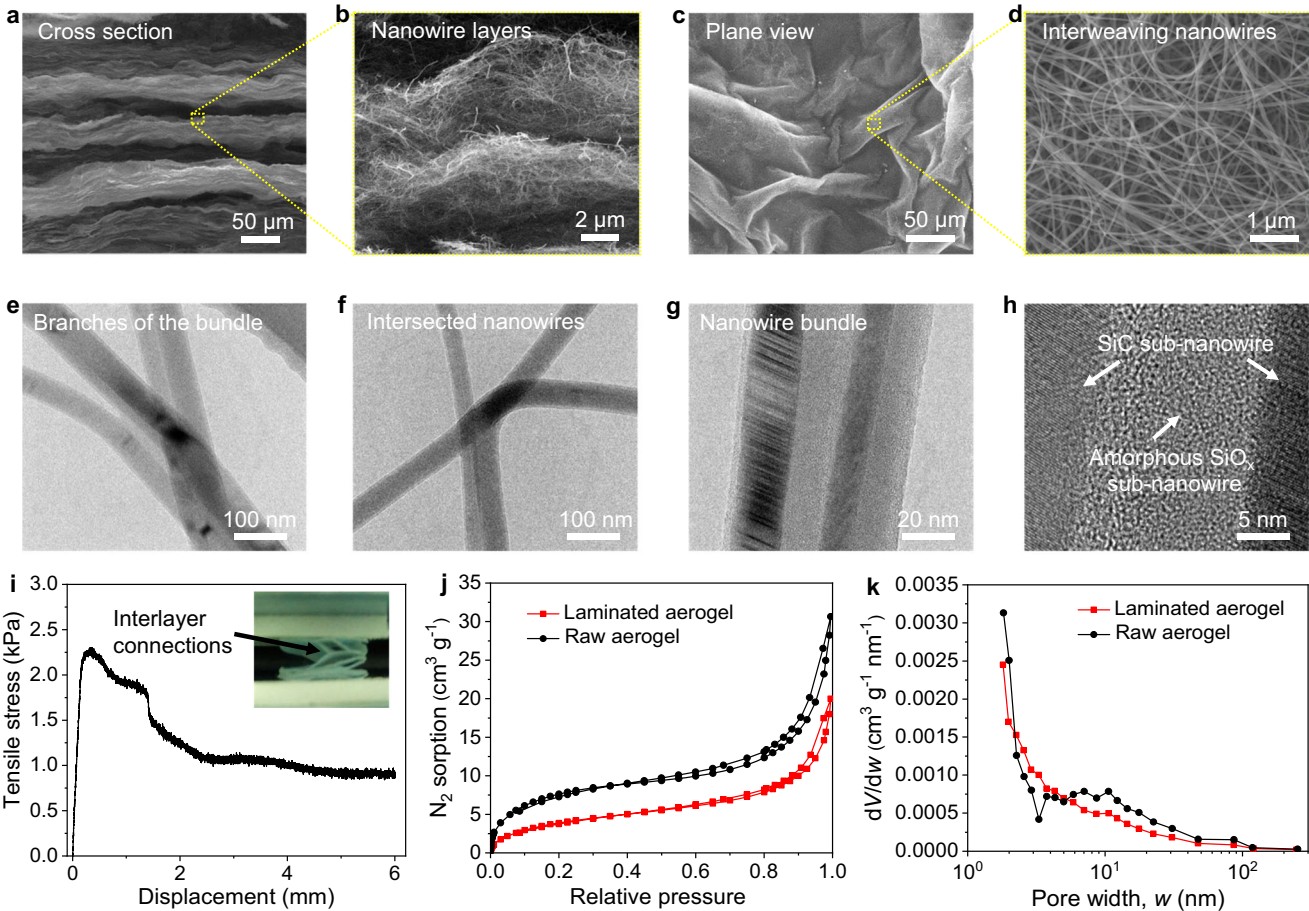

**Fig. 2 | Microstructure of the laminated SiC-SiO$_x$ nanowire aerogel. a** Cross-section morphology of the aerogel showing the laminated structure. **b** Amplified cross-section SEM image of the marked area in **a** showing the oriented SiC-SiO$_x$ nanowires in each layer. **c** Plane view of the aerogel showing the wrinkled structures in each nanowire layer. **d** Amplified SEM images of the marked area in **c** showing the well-interconnected SiC-SiO$_x$ nanowires and bundles assembled nanowire layer. **e**, **f**, **g** TEM images showing the branches of the nanowire bundles, the intersected nanowires and nanowire bundles, respectively. **h** High resolution TEM image showing the atomic level connection between the nanowires in the nanowire bundle through amorphous SiO$_x$. **i** Tensile stress-displacement curve of the aerogel along the direction perpendicular to the nanowire layers, showing the adhesion stress between neighboring layers. Insets showing the macroscopic morphology of the aerogel during the test. **j** N$_2$ sorption isotherms at 77 K and **k** pore size distribution derived from BJH analysis, respectively. $V$ is pore volume, and $w$ is pore width.

indicates that the ethanol flows along the radial direction of the aerogel paper, resulting in a radial capillary flow. This is because the liquid flow is more likely to move towards the solvent-substrate-air line[31–34]. Under the action of the radial capillary flow, the nanowires were gradually aligned along the direction of the liquid flow (Fig. 1d), which could be confirmed by the transverse wrinkles that appeared in the cross-section during the evaporation of the ethanol (Fig. 1f and Supplementary Movie 2). This kind of capillary flow-induced reorientation behavior was also observed in a tungsten oxide nanowire system during drying[35]. Simultaneously, with the evaporation of the ethanol, severe shrinkage of the aerogel in the vertical direction, resulted in the densification and further reorientation of the aerogel (Fig. 1d, f). The shrinkage of the aerogel is to compensate for the ethanol loss caused by the evaporation at the solvent-substrate-air line[35]. Therefore, it could be concluded that the radial capillary flow and vertical shrinkage of the aerogel result in the reorientation of the nanowires along the radial direction and thus the formation of the laminated microstructure.

The density of the aerogel is measured to be about 50 mg cm$^{-3}$ and the porosity is estimated to be ~98%. The increase of the density is due to the densification process during the preparation process. The laminated microstructure of the aerogel can be further confirmed by the scanning electron microscopy (SEM) analysis displayed in Fig. 2a,

which shows a significant difference with the randomly distributed nanowires in the microstructure of the raw aerogel paper (Supplementary Fig. 1) and other previously reported ceramic nanowires aerogels[12,13,15]. The layered structures at two size scales were observed. One with tens of micrometers is inherited from the raw aerogel paper (Fig. 2a), and the other is the waving sub-micrometer or even nanometer-sized nanowire layer originated from the capillary force-induced self-assembly of the nanowires (Fig. 2b). In the plane view, wrinkles with sizes ranging from several tens to more than one hundred micrometers were observed (Fig. 2c). The wrinkled layers are composed of interweaving nanowires and nanowire bundles (Fig. 2d), showing a denser packing of the nanowires than that in the raw aerogel paper (Supplementary Fig. 1). The nanowire bundles are interconnected with each other through branching and intersected nanowires (Fig. 1e, f). Transmission electron microscopy (TEM) analysis showed that the SiC-SiO$_x$ nanowires in the bundles are well-bonded with each other through amorphous SiO$_x$ (Fig. 1g, h). Altogether, a laminated ceramic aerogel with a hierarchical structure was obtained through a simple capillary force-induced self-assembly strategy.

We characterized the interlayer adhesion by applying tensile stress perpendicular to the nanowire layers. The result in Fig. 2i shows that the maximum adhesion stress is ~2.3 kPa. Even though the aerogel was delaminated during the test, some local parts between the

neighboring layers were connected with each other. To insights into the microscale interconnection between neighboring layers, we observed the interface during the delamination process. As shown in Supplementary Fig. 2, during the delamination, bridging behaviors of the nanowires were seen clearly at the interface between the neighboring laminates, showing a Velcro-like connection. Such kind of connection is formed during the sample preparation process. When two pieces of the paper-like raw nanowire aerogel connected with each other, the nanowires in one layer would be embedded in the pores in the other layer. During the evaporation of the solvent, the shrinkage of the raw nanowire aerogel took place. Under the action of the capillary force, the embedded nanowires were gradually trapped in the pores and fastened by the nanowires around the pores gradually. Therefore, after the evaporation of the solvents, a large number of nanowire-to-nanowire junctions formed at the interface. During the delamination, the nanowire-to-nanowire friction thus prevents the relative motion of the nanowires and nanowire layers, forming a good interconnection.

Figure 2j, k show the $N_2$ sorption isotherms at 77 K and pore size distribution derived from Barrett–Joyner–Halenda (BJH) analysis, respectively. The calculated specific surface areas of the laminated aerogel and raw aerogel are 15.4 and 29.1 $m^2 g^{-1}$, respectively. The average sizes of the mesopores are 11.8 and 14.3 nm, respectively. Both the specific surface area and pore size of the laminated aerogel show decreases when compared with those of the raw aerogel. The decrease in the specific surface area is related to the severe volume shrinkage of the raw aerogel during the preparation process, which results from the more compact interconnection between neighboring nanowires. Given that the $N_2$ molecule can exclusively detect the microporous and mesoporous structures, the average pore sizes calculated using the BJH method may not encompass the entire range of pore sizes. To gain a comprehensive understanding of the overall pore volume and size of the samples, we then analyzed the pore distribution in the laminated aerogel by combining the $N_2$ sorption results with mercury intrusion porosimetry and total pore calculation[36]. As discussed in the Supplementary Discussion, micropore and mesopore volumes account for only 0.15% of the total pore volume, most of the pores are macropores with a size smaller than 355 μm, and the average size of all the pores is ~5.1 μm.

## Mechanical robustness and flexibility

We then investigated the mechanical properties of the laminated aerogel. The compressive stress-strain curve was obtained at a loading rate of 0.5 mm min$^{-1}$ by using a universal tester. As shown in Fig. 3a and Supplementary Fig. 3a, the aerogel can be compressed to 20% and 40% strain consecutively, and then recover to almost its original size, demonstrating good reversible compressibility. When the compressive strains reach 60% and 80%, only small permanent deformations of ~5% and ~6% are observed, respectively. Similar to some other ultralight ceramic aerogels[13,14,30], the laminated aerogel exhibits three deformation stages during the loading process, including an initial linear stage below 40% compressive strain, a transition region with gradually increased compressive stress and stress growth rate between 40% and 60% strain, and a densification stage with rapidly increasing stress after 60% strain. The average compressive modulus calculated from the initial linear stage is about 222 ± 32.7 kPa, and the average maximum stress at 80% strain is 1255 ± 116.3 kPa, which are about 5 to more than 10 times higher than those of the previously reported resilient ceramic aerogel[5,13,14,37,38] (Fig. 3b), demonstrating the high stiffness and load-bearing capacity of the laminated SiC-SiO$_x$ nanowire aerogel.

The laminated aerogel also exhibits good fatigue resistance under compression. Figure 3c and Supplementary Fig. 3b show the stress-strain curve of the aerogel during 100 cyclic fatigue tests at a set strain of 40%. During the first 10 cycles, there is rarely permanent deformation. When the aerogel is compressed for 20 cycles, the permanent

strain is about 6%. Even though the compression cycle reaches 100, the permanent strain is only ~10%. The maximum stress at 40% strain shows a decrease at the first 20 cycles, but it keeps almost constant after then and retains up to 85% of the initial maximum stress (Fig. 3d), showing robustness under mechanical impact. During the loading-unloading process, the aerogel exhibits a high mechanical energy loss coefficient of 0.62 in the first cycle and keeps constant at around 0.30 after 20 cycles, comparable to other resilient ceramic aerogels[13,14], indicating the efficient dissipation of mechanical energy and thus good mechanical impact resistance.

Besides the attractive compressive properties, the laminated aerogel also shows robustness and flexibility under tensile stress. Figure 3e displays the tensile stress-strain curve of the laminated aerogel, showing a four-stage deformation behavior. The deformation starts with an initial linear region below 1.2% strain, exhibiting an average tensile modulus of 4855 ± 111.0 kPa. Between 1.2% and 6.0% strain, it is a transition region with gradually increased stress but decreased growth rate (The mechanisms for such transition are discussed in the Deformation mechanism part). After 6.0% strain, the stress rapidly increases with the increase of strain, showing a large-strain ductile deformation behavior (Supplementary Movie 3 and insets in Fig. 3e). When the tensile strain reaches up to 20% strain, an average maximum tensile stress (fracture strength) of 399 ± 83.4 kPa is observed. Then the stress gradually decreases to 0 until about 22% strain, showing a nonbrittle fracture. Such fracture behavior is related to the gradual initiation and propagation of the crack in the laminated aerogel. As shown in Supplementary Fig. 4, during the propagation, the crack shows six deflections before the break. After the break, the sample still exhibits its integrality (insets in Fig. 3e and Supplementary Fig. 5). The strength and modulus are up to 10 and 20 times higher than that of our previously reported stretchable SiC-SiO$_x$ nanowire aerogel (with a strength of 30 kPa and a modulus of 204 kPa)[30], respectively, but the fracture strain rarely decreases, verifying the superiority of the laminated microstructure design. The tensile strength is 5 to more than 10 times higher than those of currently reported highly stretchable ceramic aerogels (Fig. 3f)[17,30,39,40]. To our knowledge, this is the highest tensile strength reported for resilient ceramic aerogel so far.

Benefiting from the mechanical robustness and large-strain deformability under both compressive and tensile deformation, the aerogel exhibits attractive performance under bending in which a complex compressive-tensile stress state exists. Figure 3g shows the bending stress-displacement curve of the laminated ceramic aerogel obtained by using a three-point bending test. To evaluate the reversible bendability of the aerogel, we introduced four consecutive loading-unloading cycles at the displacement of 5.5, 6, 6.5, and 7 mm, respectively. As shown in Fig. 3g and its insets, the laminated aerogel displays recoverable bendability except for some permanent deformation. When the displacement reaches 9 mm, an average bending strength of 261 ± 11.4 kPa is observed. After 9 mm displacement, the sample is gradually trapped in the 3-point testing clamp, forming an angle of 90° between the two sides. Further increasing the displacement to 15 mm, the angle decreases to less than 45° (Supplementary Fig. 6a), however, no visible damage is observed in the sample, displaying the flexibility of the laminated aerogel under bending deformation. When taking down the sample from the clamp after the test, the angle recovers to about 130° (Supplementary Fig. 6b), showing good recoverability, indicating an elastoplastic deformation during bending. Such deformation behavior corresponds well to the elastic compressibility and ductile tensile deformation of the laminated aerogel as illustrated above.

We further used a 2-point buckling test to investigate the flexibility of the laminated aerogel under complex compressive and tensile stress states. As shown in Fig. 3h and its insets as well as Supplementary Movie 4, the aerogel can deform to 80% buckling strain and recover to almost its original shape. The maximum buckling stress is about

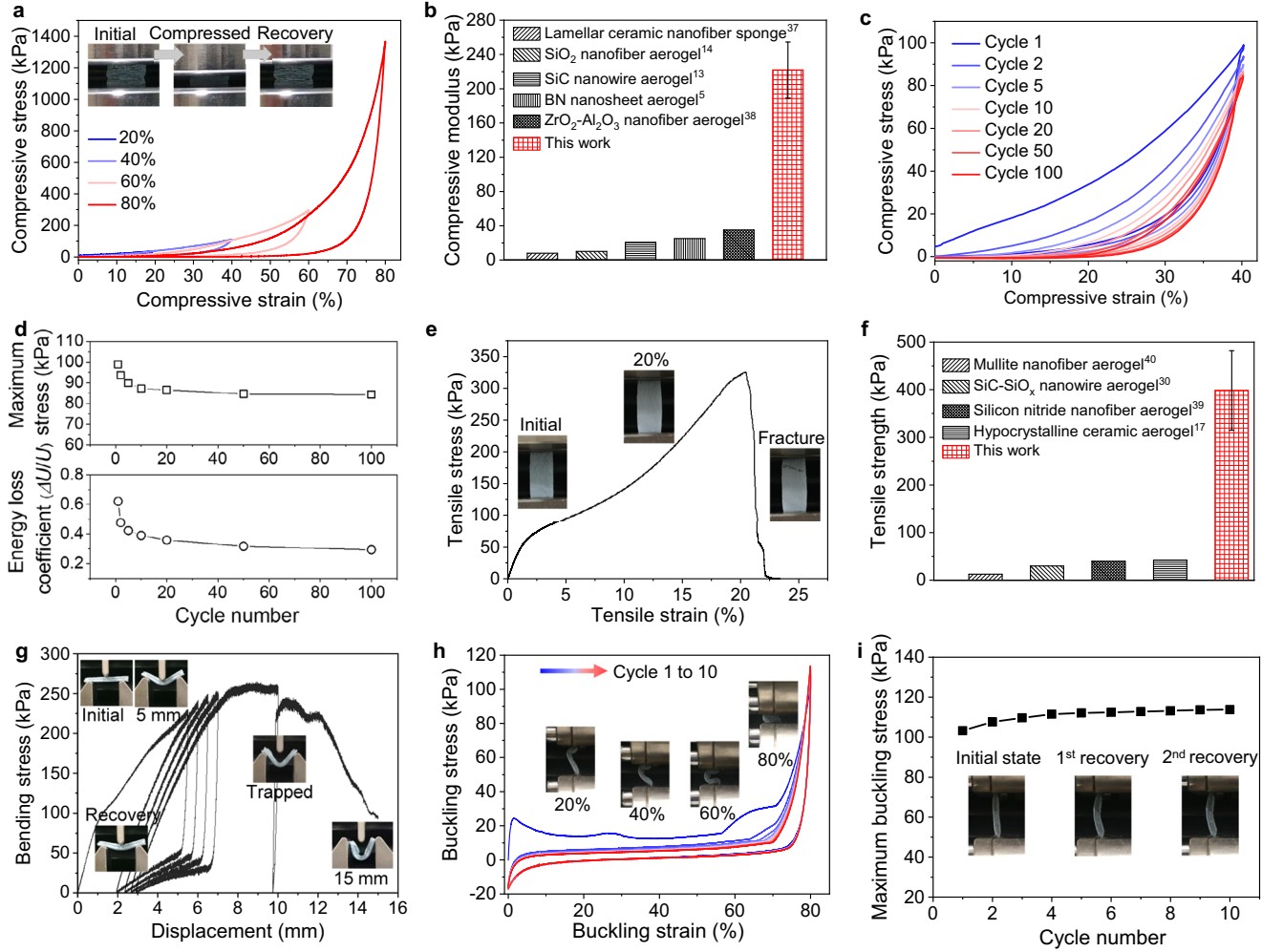

**Fig. 3 | Mechanical properties of the laminated SiC nanowire aerogel.**
**a** Consecutive compressive stress-strain curves from 20% to 80% strain, showing reversible compressibility and high compressive stress. Insets showing the recoverable compressibility. **b** Comparison of the compressive modulus of the laminated aerogel and other resilient ceramic aerogels[5,13,14,37,38]. **c** Cyclic compressive fatigue test at set strain of 40%, indicating good reversibility and fatigue resistance. **d** Evolution of the maximum compressive stress and mechanical energy loss coefficient during the 100 cyclic fatigue test. ΔU is the dissipated energy during loading–unloading process, and U is the work done during the loading process. **e** Tensile stress-strain curve of the laminated aerogel, showing the high strength, high modulus and ductile deformation behavior. **f** Comparison of the tensile

strength of the laminated aerogel with other stretchable ceramic aerogels[17,30,39,40]. **g** Bending stress-displacement curve of the laminated aerogel, showing partially reversible and highly bendability. Insets showing the macroscopic morphology evolution during the loading-unloading processes. **h** 10 cycles fatigue buckling deformation test from 0 to 80% buckling strain. Insets showing the morphology of the aerogel at buckling strain of 20%, 40%, 60%, and 80%, respectively. **i** Evolution of the maximum buckling stress of the laminated aerogel during a cyclic fatigue test. Insets showing the slight permanent deformation of the laminated aerogel after recovery from the 80% buckling deformation. Error bar in **b**, **f** standard deviations derived from three performed measurements.

103 kPa at 80% strain, which is up to 20 times higher than previously reported nanofibrous aerogels[16,18]. Such buckling-recovery is also repeatable without a decrease of the maximum buckling stress (Fig. 3h, i), showing good fatigue resistance under a complex stress state. Worth noting that after the buckling deformation, there is slight permanent deformation observed in the sample, as evidenced by the appearance of tensile stress (negative part) in the buckling stress-strain curve (Fig. 3h) and the bending deformation in the recovered sample (insets in Fig. 3i). These phenomena indicate the laminated aerogel could withstand accidental deformation during services instead of abrupt failure as observed in conventional ceramic aerogels.

## Deformation mechanisms

To understand the deformation mechanisms, in situ compressive and tensile tests were conducted in SEM. As shown in Fig. 4a, with the increase of compressive strain, the SiC nanowire layers were compressed to be thinner and thinner, and the waving layers became

straighter. During this process, the nanowire layers show a slight bending deformation behavior but large-displacement synergistic movement (Fig. 4b), which is notably different from the large buckling and bending deformation of the nanowires in the aerogel with random structures[12,13,23,28]. Such difference is attributed to the more severe constraint around individual nanowires and between neighboring layers in the laminated structure, which could increase the deformation resistance of the nanowire and thus the modulus and strength of the aerogel. Despite the increasing deformation resistance, the high porosity (~98%) of the laminated aerogel could still provide enough moving room for the nanowires. Therefore, the compressive strain could be still very large. During the unloading process, benefiting from the flexibility of the nanowires, the nanowire layers can recover to almost their initial states (Supplementary Movie 5 and Fig. 4a).

During the tension, the deformation mainly takes place at the wrinkles. As shown in Fig. 4c, with the increase of the tensile strain, the concave-convex surface in nanowire layers was stretched to be more

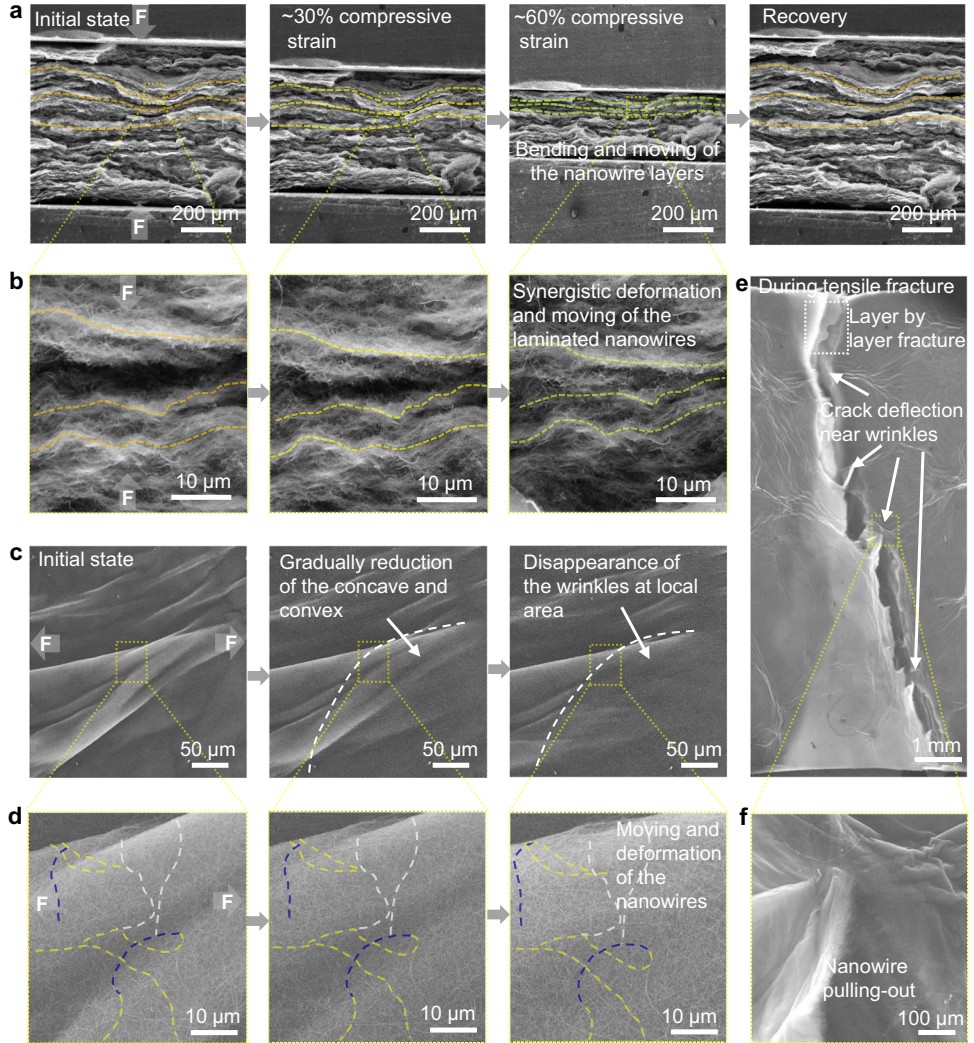

**Fig. 4 | Deformation mechanisms of the laminated aerogel under compression and tension. a** Microstructure evolution of the nanowire layers observed from the cross-section during the compression and recovery. **b** Synergistic deformation and moving of the nanowire layers during compression. **c** Evolution of the plane view morphology of the aerogel during tensile deformation. **d** The transformation from a concave-convex surface to a flat surface of a wrinkle through the synergistic deformation and moving of the laminated nanowire. **e** Crack deflection taking place at the wrinkles during fracture. **f** The amplified microstructure in the inset in **e** shows the pulling-out and nanowire bridging behavior during the crack propagation.

and more smooth, resulting in the gradual disappearance of wrinkles and thus the large permanent deformation of the aerogel. When we focused on the individual nanowires, it could be seen that, (i) the nanowires in the concave region were stretched and moved synergistically (marked by the yellow dash lines in Fig. 4d), resulting in the decrease of the concave, (ii) when a concave transforms into a convex, the nanowires in the concave were further bent (marked by the blue dash line in Fig. 4d), (iii) most of the nanowires in the flat area showed rarely deformation but a synergistic movement (the nanowires marked by the white dash line in Fig. 4d is shown as examples). Such deformation and movement behaviors are very different from the large-strain bending or buckling deformation of the nanowires as observed in the aerogel with randomly distributed nanowires[30], which is attributed to the increased friction between neighboring nanowires and thus the increased deformation and movement resistance of the individual nanowire generated from their denser distribution. The wrinkled morphology of the nanowire layers could increase the friction between contacted nanowire layers during the tensile deformation due to the rough interface, while the disappearance of the wrinkles during the deformation can increase the tensile strain of the aerogel. The transition in the stress growth rate between 1.2% and 6%

tensile strain is related to the evolution of the wrinkles in the aerogel during tensile deformation. With the increase of the tensile strain, the wrinkles were gradually stretched, resulting in the gradually decreased interaction between neighboring layers, and thus the decreased stress growth rate. Altogether, it is concluded that the intensive interaction among individual nanowires and between neighboring nanowire layers are responsible for the high tensile strength and modulus, while the flexibility of the nanowires and gradual disappearance of the wrinkles are the origins of the large-strain tensile deformation.

We also observed the morphology of the laminated nanowire aerogel during the tensile fracture. Figure 4e, f shows the intense crack deflections and nanowire pulling-out behavior, respectively, corresponding well to the observations in Supplementary Fig. 4. Notably, the crack deflection usually takes place where the wrinkle exists (Fig. 4e). To insight into such fracture mechanism, we also recorded the evolution of the nanowire morphologies at the wrinkles before fracture. As shown in Supplementary Fig. 7, in some local areas at the wrinkle, the nanowires were stretched to be more and more straight, forming local nanowire bridges. The nanowire bridges and the fold lines at the wrinkle promote and guide the deflection of the crack (Fig. 4e). Moreover, the laminated microstructure also

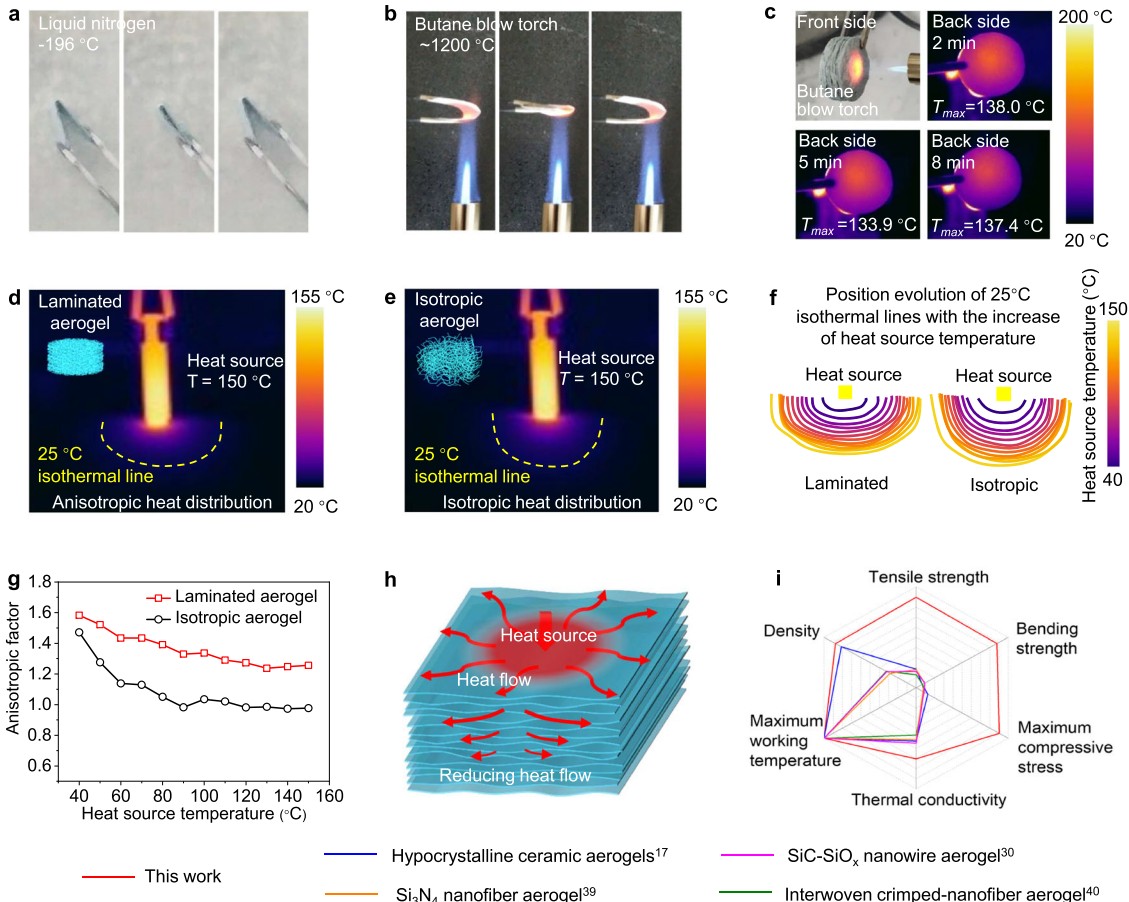

**Fig. 5 | Thermal stability and thermal insulation of the laminated aerogel.**
**a**, **b** The reversible bendability of the aerogel in liquid nitrogen and under a butane blow torch, respectively. **c** Infrared images showing the distribution of temperature on the backside of aerogel, illustrating the thermal insulation performance of the laminated aerogel under the heating of a butane blow torch. **d**, **e** Infrared images showing the shapes of 25 °C isothermal lines in the laminated aerogel and the isotropic aerogel, respectively, indicating the anisotropic and isotropic thermal conducting behavior, respectively. **f** Position evolution of the 25 °C isothermal line with the increase of heat source temperature during the heating process from room temperature to 150 °C of the laminated aerogel and isotropic aerogel. **g** The anisotropic factor of the shape of 25 °C isothermal lines in the laminated aerogel and isotropic aerogel. **h** Thermal insulation mechanism of the laminated aerogel. **i** Comparison of the thermal and mechanical properties of the laminated aerogel and other highly stretchable, bendable, and compressible ceramic aerogels[17,30,39,40], indicating a similar thermal property but much higher strength under tension, bending, and compression.

provides a layer-by-layer fracture behavior of the aerogel (as evidenced by the marked white dash box in Fig. 4e and Supplementary Fig. 5c). These phenomena work together to lead to increase of crack propagation path and thus nonbrittle fracture of the laminated aerogels.

Based on the above observations and discussions, it is seen that the increase in the strength and modulus of the laminated aerogel can be attributed to the enhanced constraints for the deformation of the nanowires during the compressive and tensile deformations. In the ceramic nanowire-based aerogels with random structure, the bending and buckling deformation of the flexible nanowires provide the aerogel large deformation capacity, the pores provide the deformation room for the nanowires[13,30], which are the precondition for the large deformation of the nanowire aerogel. The interconnection joints between the nanowires play the role of load transferring points, enabling the synergistic deformation and recovery of the nanowires network through constraining the deformation of the nanowires[13,23]. The modulus and strength of the aerogels are determined by the deforming resistance of the nanowire provided by the constraints. In the present case, the laminated layers provide planar constraints to the deformation of the nanowires, resulting in an increase in the deformation resistance. The dense distribution of the nanowires also results in intense interaction among individual nanowires. Therefore, the

bending and buckling resistances of the nanowires in the laminated aerogel are highly increased, thus resulting in high modulus and strength. Moreover, the Velcro-like connection between neighboring layers provides the nanowires and nanowire layers good flexibility under bending deformation (as verified by the waving behavior of the nanowire layers in the bent sample in Supplementary Fig. 8), thus resulting in good flexibility under bending and buckling. These results suggest that the key to further optimizing the mechanical properties of the ceramic aerogel is to tune the deforming resistance of the building blocks in their microstructures.

## Thermal stability and thermal insulation

We then evaluated the thermal stability of the laminated aerogel. As shown in Fig. 5a and Supplementary Movie 6, the aerogel could maintain its reversible bendability at −196 °C in liquid nitrogen. The reversible bendability also retained under the flame of a butane blow torch (with a high temperature of -1200 °C) (Fig. 5b and Supplementary Movie 7). These results show its good mechanical flexibility and thermal stability in a broad temperature range, which can be attributed to the ceramic nature of the laminated SiC-SiO$_x$ nanowire aerogel. However, it should be noted that to further reflect the mechanical properties of the aerogel under such working temperature needs more precious characterization.

We evaluated the thermal insulation performance by using a butane blow torch as a high-temperature heating source. A thermal imager was used to record the temperature distribution on the backside of a piece of laminated aerogel with a thickness of 10 mm. Although the maximum temperature of the butane blow torch can reach as high as more than 1200 °C, the stabilized temperature on the backside is only ~135 °C (Fig. 5c), indicating that the laminated aerogel is an attractive thermal insulator.

The measured result shows that the laminated SiC nanowire aerogel exhibits a low room temperature thermal conductivity of ~39.3 ± 0.4 mW m$^{-1}$ K$^{-1}$) in the direction perpendicular to the laminated structure (Supplementary Table 1), which is comparable to that of the silk cocoons[27,28], showing good thermal insulation performance. As is known, solid conduction, gas conduction, gas convection, and radiation are the factors that influence the thermal conductivity of porous materials. The contribution of radiation is negligible at room temperature. Gas convection is also inhibited in porous material with pore sizes less than 4 mm[41,42]. The measured result (Fig. 2k and Supplementary Discussion) shows that the pore size is less than 355 μm in the laminated aerogel, indicating there is rarely gas convection. Therefore, the thermal conductivity of the laminated aerogel can be attributed to the solid conduction and gas conduction.

Solid conduction of the laminated SiC-SiO$_x$ nanowire aerogel depends on the phonon conduction. The high porosity (~98%) can block and the nanowires-assembled tortuous structure can prolong the phonon conducting pathway, thus reducing solid conduction. Moreover, the phonon conduction barriers caused by the amorphous SiO$_x$ sub-nanowire and the stacking faults in the SiC sub-nanowire can further reduce solid conduction[13,30,43–46]. These factors are responsible for the decrease in thermal conductivity. For comparison, the density of the laminated aerogel (50 mg cm$^{-3}$) is about nine times higher than that of the raw SiC-SiO$_x$ nanowire aerogels with random microstructure (with a density of 5.7 mg cm$^{-3}$ and thermal conductivity of 28.4 mW m$^{-1}$ K$^{-1}$)[30], which indicates that the solid conduction in the laminated aerogel might be nine times higher than that of the raw SiC-SiO$_x$ nanowire aerogels. However, the thermal conductivity only shows an increase factor of ~1.37. Therefore, there are other factors that are responsible for good thermal insulation performance.

Compared with the pore size distribution in the raw SiC-SiO$_x$ nanowire aerogel, in the laminated aerogel, the average width of the nanosized pores shows a further decrease. According to the Knudsen effect[42], the gas conduction reduced dramatically with the decrease of pore width, especially when the pore sizes are below the mean molecular free path of air (~70 nm). Therefore, gas conduction in the laminated aerogel is smaller than that in the raw nanowire aerogel. However, as illustrated in the Supplementary Discussion, the volume of micropores and mesopores in the laminated aerogel is only about 0.15% of the total pore volume, which indicates that the contribution from the decreased gas conduction to the thermal insulation is limited.

We then observed the heat transportation behavior in the laminated aerogel. For comparison, we also recorded the heat distribution in an isotropic SiC-SiO$_x$ nanowire aerogel. As shown in Supplementary Fig. 9, a cylinder heater with a diameter of 2 mm was used as the heat source to generate heat on the surface of the aerogel samples. It could be seen that during the heating process, the 25 °C isothermal line shows an anisotropic shape in the laminated aerogel (Fig. 5d, f, Supplementary Fig. 10 and Supplementary Movie 8), which is obviously different from the isotropic shape in the isotropic aerogel (Fig. 5e, f, Supplementary Fig. 11 and Supplementary Movie 9). We then calculated the anisotropic factor of the 25 °C isothermal lines (The ratio between the horizontal length and vertical length in the shape of the 25 °C isothermal lines). As shown in Fig. 5g and Supplementary Tables 2 and 3, with the gradual increase of the heat source temperature during the heating process, the anisotropic factor of the laminated aerogel gradually stabilizes at around 1.3, while that of the isotropic aerogel is

~1.0 (Fig. 5g). These observations indicate the anisotropic thermal conducting behavior of the laminated aerogel. Such anisotropic heat flowing behavior could increase the heat transportation along the nanowire layers, thus reducing the heat transportation and increasing the thermal insulation in the direction perpendicular to the laminated structure (Fig. 5h). Therefore, the good thermal insulation of the laminated aerogel is attributed to not only the high porosity, and the phonon conducting barriers but also the decreased width of nanosized pores and the laminated microstructure induced anisotropic heat transport behavior.

Benefiting from its special microstructures, the laminated aerogel thus represents one of the strongest resilient ceramic aerogels with strengths and modulus of up to an order of magnitude larger than previously reported ceramic aerogels[17,30,39,40] under various deformations (including compressive, tensile and bending deformations) but without compromising its flexibility and thermal insulation significantly (Fig. 5i).

In summary, we have demonstrated a laminated microstructure design strategy in SiC-SiO$_x$ nanowire aerogel to resolve the conflicts among mechanical robustness, deformability, and thermal insulation in ceramic aerogels. Such rational structure design not only enhances the deformation and moving resistance of the nanowires but also keeps their flexibility. It also produces an anisotropic thermal conducting behavior to reduce heat flux transportation along the direction perpendicular to the laminated layers. These combined mechanical and thermal properties make the aerogel a strong yet flexible thermal insulator for application in conditions where compressive, tensile, bending, and buckling deformations would take place such as dynamic thermal sealing materials for aerospace vehicles and thermal insulation materials for lunar rovers and abyssal bathyscaphe.

## Methods

### Preparation of the raw material

The preparation of SiC-SiO$_x$ nanowire aerogel paper includes two main steps, one is the synthesis of a siloxane xerogel and the other is the decomposition of the xerogel in Argon. Methyltrimethoxysilane (MTMS, 99% purity, Meryer (Shanghai) Chemical Technology Co., Ltd., China) and dimethyldimethoxysilane (DMDMS, 99% purity, Meryer (Shanghai) Chemical Technology Co., Ltd., China,) were used as the raw materials for the synthesis of siloxane xerogel. The weight ratio of MTMS and DMDMS was 1:4. During the synthesis, ethanol (≥99.7% purity, Sinopharm Chemical Reagent Co., Ltd., China), deionized water, and nitric acid (65.0–68.0% concentration, Sinopharm Chemical Reagent Co., Ltd., China) were used as the solvent, hydrolytic reagent, and catalyst, respectively. The mixture of these materials was stirred and then let it stand to form a siloxane gel. The gel was then dried at 100 °C for 2 h to obtain the siloxane xerogel. The xerogel was then loaded in a graphite crucible in a gas-pressure furnace. During the decomposition of the siloxane xerogel in argon with a pressure of 0.25 MPa at 1150 °C for 3 h, the decomposed SiO and CO gases reacted with each other, resulting in the growth and self-assembly of SiC-SiO$_x$ nanowires in the crucible. After the decomposition, a SiC-SiO$_x$ nanowire aerogel paper formed on the inner surface of the crucible.

### Preparation of the laminated aerogel

The obtained SiC-SiO$_x$ nanowire aerogel paper was cut into pieces. The pieces were then stacked into a bulk aerogel. The bulk aerogel was then immersed into ethanol, after it was fully infiltrated, it was put out and then dried naturally. After the drying, the laminated SiC-SiO$_x$ nanowire aerogel was obtained. To illustrate the formation process of the laminated aerogel, we observed the microstructure evolution of a piece of raw paper-like aerogel during the evaporation of ethanol by using an Olympus BX51 optical microscope. We attached a piece of the paper-like aerogel on a glass slide mounted on a heating element with a temperature of 50 °C. Then we dropped ethanol on the aerogel paper

and observed the microstructure evolution during the ethanol evaporation process.

## Materials characterization

The density of the laminated aerogel was calculated by using the mass and volume of a piece of the aerogel. The porosity was estimated by using the density of amorphous silica ($2.2\,g\,cm^{-3}$) and 3C-SiC ($3.2\,g\,cm^{-3}$). The microstructures of the laminated aerogel and the nanowires are imaged by an SEM (Quanta 600, FEI, USA) and a field emission TEM (JEM-2100, JEOL, Japan). The interlayer bonding stress between neighboring laminates was analyzed by testing the tensile stress in the direction perpendicular to the laminates. The interface during the delamination of the neighboring layers was analyzed by SEM. Pore size distribution in the aerogel was analyzed by integrating the $N_2$ sorption (ASAP2020, Micromeritics, USA), mercury intrusion porosimetry (AutoPore IV 9500, Micromeritics, USA), and the total pore calculation[36]. The detailed analysis of the pore size distribution is displayed in the Supplementary Discussion.

## Mechanical properties

The compressive, tensile, and bending stress-strain curves of the laminated aerogel are tested by using a universal tester (Suns Co., Ltd., China) (load cell:1000 N) at a loading-releasing rate of $0.5\,mm\,min^{-1}$. The compressive modulus was calculated from the initial linear deformation region in the compressive stress-strain curve from 0 to 40% strain. The area of the hysteresis loop in the loading-unloading compressive curves is the dissipated energy, while the area below the loading curve represents the compressive work. The energy loss coefficient is equal to the ratio of the dissipated energy and the compressive work. The tensile modulus was calculated from the initial linear deformation region in the tensile stress-strain curve below 1.2% strain. The average values and standard deviations of the modulus and strength were calculated based on the values of three samples. The related data is shown in Supplementary Table 4. The buckling stress-strain curves were obtained at a loading-releasing rate of $10\,mm\,min^{-1}$. In situ SEM compressive and tensile experiments of the laminated aerogel were conducted by using a test unit (Microtest 4000 N, Deben, Britain).

## Thermal insulation

We measured the thermal conductivity of the laminated aerogel by using a transient hot-wire method (XIATECH, TC3000E, China). The measured data is shown in Supplementary Table 1. The thermal conductivity was measured 5 times. We used the averaged value as the thermal conductivity of the laminated aerogel. The infrared images during the heating process were recorded by using a thermal infrared imager (FOTRIC 285, China). The detailed experimental setup was illustrated in Supplementary Fig. 9. The analysis of the position of the isothermal lines was performed by using the FOTRIC AnalyzIR system. The calculation of the anisotropic factor of the shape of the isothermal lines is illustrated in Supplementary Fig. 12 and the measured data is displayed in Supplementary Table 2 and Supplementary Table 3.

# Data availability

The data that supports the findings of this study are available in the article and supplementary information file. Source Data file has been deposited in Figshare under accession code https://doi.org/10.6084/m9.figshare.24088536.

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

## Acknowledgements

The authors thank the valuable discussions with Prof. Hu Zhang and Dr. Shugui Yang from Xi'an Jiaotong University. We also thank the financial support from the National Natural Science Foundation of China (52102076, 92263204), the China Postdoctoral Science Foundation (2021M690122), the Top Young Talents Program of Xi'an Jiaotong University, and the Fundamental Research Funds for the Central Universities.

## Author contributions

H.W. and S.J. supervised the study. L.S. conceived and designed the research. L.S. prepared the sample and did the characterization and analysis with the help of X.L., J.R., S.-W.G., P.G., Z.C., D.L., M.N., L.Z. and K.P. L.S. prepared the manuscript. All the authors discussed the results and commented on the manuscript.

## Competing interests

The authors declare no competing interests.
