## [Peer Review File · Nature Communications]

Strong yet Flexible Ceramic AerogelReviewer #1 (Remarks to the Author):

This paper proposed a novel preparation of ceramic aerogel that displays not only flexibility under compressive, tensile and buckling deformations but also high strength and modulus of up to an order of magnitude larger than other ceramic aerogels, which represents a great advance in the development of resilient ceramic aerogels. The idea is exciting and the procedure was clearly described, however, there are several concerns need to be clarified.

1. In the part of "Mechanical robustness and flexibility under complex stress states", the authors show the energy loss coefficient of the aerogel during compressive, but the method used for the calculation of the coefficient did not mentioned, which should be added in the Methods part.
2. The laminated ceramic aerogel is anisotropic, hence the thermal and mechanical properties should also be different in different orientations, probably authors could provide more information to show such effect or any pros and cons of such design.
3. During the tensile deformation, the authors described that "Between 1.2% and 6.0% strain, it is a transition region with gradually increased stress but decreased growth rate". Could the author explain the reason for such transition?
4. In the sentence "As shown in Supplementary Fig. S, in some local areas at the wrinkle, the nanowires were stretched to be more and more straight, forming local nanowire bridges", "Supplementary Fig. S" should be "Supplementary Fig. 5".
5. The sentence "The laminated aerogel thus represents one of the strongest resilient ceramic aerogels with strengths and modulus of up to an order of magnitude larger than previously reported ceramic aerogels under complex stress states (compressive, tensile and bending) but without compromising its flexibility and thermal insulation significantly (Fig. 4i)" with citations, should be moved to the paragraph above the conclusion part. And the related expression in the conclusion part should revised.
6. All the chemicals used in the paper, e.g. methyltrimethoxysilane and dimethyldimethoxysilane, their information (concentration, provider etc.) should be provided.

Reviewer #2 (Remarks to the Author):

The authors reported a set of properties of laminated aerogel materials obtained by stacking pieces of nanowire aerogel under ethanol. The original premise that the resultant materials mimics silk cocoon structure is a significant stretch. The authors did not provide supporting data that the layers of aerogels had significant "bonding" to make the laminates any different from mere stacking the individual layers. The work in its current form lacks sufficient novelty. Some specific comments are listed below.

1. The authors used the word "extreme" as several places in reference to temperature and stress level without defining for the sake of readers what this term really means. I suggest that either the authors refrain from using such a term or provide a clear definition. A similar issue is found with the term "ultralow".
2. p.3, "It shows a high compressive modulus of 230 kPa, high compressive stress of 1366 kPa at 80% strain, high tensile stress of 325 kPa and modulus of 4.65 MPa, high bending strength of 260 kPa, which are several to tens of times higher than other resilient ceramic aerogel." These values do not contain measurement errors, such as standard deviation values. These should be reported.
3. The sentence quoted in #2 above does not present the work in good stead. Are the above properties enough? A factor of "several to tens of times higher" of low numbers may be well-below what is expected. I ask the authors to list a set of representative target properties here. In my view, porous aerogel materials with 98% porosity as in this work would never be used in load bearing applications. Instead, the question should be if the aerogel products can withstand handling stress? In this context, the authors should define a range of handling stresses.

4. p.3, "The flexibility also maintains under 1200 °C butane blow torch and at temperature as low as -196 °C in liquid nitrogen." The statement/claim is not convincing without some real stress vs. strain data.

5. The raw SiC-SiO_x nanowire aerogel was reported in a previous publication, Ref. 28. This presents a question on the novelty of the present contribution.

6. The process of fabrication of laminated aerogel is too simplistic. Ethanol evaporation from the pores brings out "self-assembly" of the particles through capillarity. It is a common knowledge that ethanol evaporation leads to densification of the aerogel. Authors needs to elaborate the physics of assembly process.

7. The difference of images in Figure 1(d) and (f) is not clear - the former uses a scale bar of 2 micrometer, while the latter with 1 micrometer. Yet, the figure in 1(f) is a lot more revealing and does not show broken nanowires, as in 1(d). Why?

8. p.5, "As shown in Fig. 2a, the aerogel can be compressed to 20%, 40% and 60% strain consecutively, and then recovers to almost its original size, demonstrating the good reversible compressibility." This claim cannot be true. Figure 2(a) shows hysteresis, although authors did not clearly indicate compression part on the figure. Thus, these materials underwent significant structural damage. The inset in this figure without a scale bar cannot clearly support the claim. I ask authors to present stress-strain diagram at low strain (less than 10%). Also, how much time does the shape recovery take?

9. What is missing from this work is a test to evaluate the interlayer "adhesion" that the authors claimed to have achieved. In my view, the properties that the authors reported in this work would be shown by individual layers of the paper.

10. I do not see any explanation of heat transport in these materials. No new information is presented. Readers would love to relate heat transport to pore size and its distribution.

Point-by-Point Response to the Reviewers' Comments

Title: Strong yet Flexible Ceramic Aerogel

Manuscript ID: NCOMMS-22-48377

COMMENTS TO AUTHOR:

Reviewer #1 (Remarks to the Author):

This paper proposed a novel preparation of ceramic aerogel that displays not only flexibility under compressive, tensile and buckling deformations but also high strength and modulus of up to an order of magnitude larger than other ceramic aerogels, which represents a great advance in the development of resilient ceramic aerogels. The idea is exciting and the procedure was clearly described, however, there are several concerns need to be clarified.

Thank you for your positive comments. We have revised the manuscript according to your suggestions. The details are shown below.

1. In the part of “Mechanical robustness and flexibility under complex stress states”, the authors show the energy loss coefficient of the aerogel during compressive, but the method used for the calculation of the coefficient did not mentioned, which should be added in the Methods part.

Response: The area of the hysteresis loop in the loading-unloading compressive curves is the dissipated energy, while the area below the loading curve represents the compressive work. The energy loss coefficient is equal to the ratio of the dissipated energy and the compressive work. We have added the details in the Methods part, as shown below.

“The area of the hysteresis loop in the loading-unloading compressive curves is the dissipated energy, while the area below the loading curve represents the compressive work. The energy loss coefficient is equal to the ratio of the dissipated energy and the compressive work.”

2. The laminated ceramic aerogel is anisotropic, hence the thermal and mechanical properties should also be different in different orientations, probably authors could provide more information to show such effect or any pros and cons of such design.

Response: Thank you for the suggestion.

1) For the mechanical property:

We tested the tensile deformation behavior along the direction perpendicular to the nanowire aligned layers. The interlay adhesive stress is only about 2.25 kPa, showing a weak conjunction between neighboring layers. This is far below the high tensile strength (398.5 ± 83.4 kPa) along the nanowire alignment direction. However, this weakness is also an advantage of the present laminated aerogel. This is because the low interlayer adhesive stress could provide the nanowires greater degrees of freedom to

deform and thus help the laminated aerogel to maintain their flexibility under tensile and bending deformations.

If the interlayer adhesion is enhanced, such flexibility cannot maintain. For example (Figure R2), in a laminated SiC@SiO₂ nanowire aerogel prepared by hot-pressing method, the interlayers bonding is largely enhanced, but the structure showed brittle fracture under bending deformation. This result further shows the advantage of the present microstructure design.

Figure R1 (Figure 2i in the revised manuscript). Tensile stress-displacement curve of the aerogel along the direction perpendicular to the nanowire layers, showing the adhesion stress between neighboring layers. Inset showing the macroscopic morphology of the aerogel during the test.

Figure R2. Microstructure of a nanowire-assembled laminated aerogel prepared by hot-pressing method. The nanowire is well bonded, exhibiting a strong interlayer bonding. However, the strong bonding will decrease the flexibility of the nanowire, thus resulting in the brittle bending fracture.

2) For the thermal insulation performance:

To illustrate the thermal insulation mechanisms, we compared the pore size distribution and thermal conducting behavior of the laminated aerogel and the raw aerogel with randomly distributed nanowire.

As shown in Figure R3, four peaks at 457 nm, 2.1 μm, 36.2 μm, and 89.5 μm were observed in pore diameter distribution curve of the laminated aerogel (Figure R3a). The volume ratio of the nanosized pores (100 nm to 1 μm) is about 2%. For comparison, in

the raw SiC-SiO_x nanowire aerogel, the pore diameters were above 4 μm with a pore diameter peak of 142.4 μm (Figure R3b). These results indicate that in the laminated aerogel the pore size shows further decrease, which can be attributed to the denser packing of the nanowires in the laminated aerogel. According to previous study, when the pore size is below 1 μm, gas conduction is reduced dramatically because of the remarkable Knudsen effect. Therefore, gas conduction in the laminated aerogel is smaller than that in the raw nanowire aerogel.

Figure R3. Pore diameter distributions in (a) the laminated aerogel and (b) the raw aerogel, respectively. The laminated aerogel showing a decrease of the pore diameter.

We also compared the different thermal conducting behavior in the laminated aerogel and isotropic aerogel. As shown in Figure R4, the laminated aerogel shows an anisotropic thermal conducting behavior with an anisotropic factor of 1.3, while that of the isotropic aerogel is about 1. Such anisotropic heat flowing behavior could increase the heat transportation along the nanowire layers, thus reducing the heat transportation and increasing the thermal insulation in the direction perpendicular to the laminated structure.

Figure R4. a, b. Infrared images showing the shapes of 25 °C isothermal lines in the

laminated aerogel and the isotropic aerogel, respectively, indicating the anisotropic and isotropic thermal conducting behavior, respectively. **c.** Position evolution of the 25 °C isothermal line with the increase of heat source temperature during the heating process from room temperature to 150 °C of the laminated aerogel and isotropic aerogel. **d.** The anisotropic factor of the shape of 25 °C isothermal lines in the laminated aerogel and isotropic aerogel.

3. During the tensile deformation, the authors described that “Between 1.2% and 6.0% strain, it is a transition region with gradually increased stress but decreased growth rate”. Could the author explain the reason for such transition?

Response: Such transition is related to the evolution of the wrinkles in the aerogel during tensile deformation. The wrinkles in the aerogel could increase the interaction between neighboring layers, thus resulting in enhanced mechanical properties. With the increase of the deformation strain, the wrinkles were gradually stretched, resulting in the gradually decreased interaction between neighboring layers, and thus the decreased stress growth rate. We have added the explanation in the deformation mechanism part.

“The transition in the stress growth rate between 1.2% and 6% tensile strain is related to the evolution of the wrinkles in the aerogel during tensile deformation. With the increase of the tensile strain, the wrinkles were gradually stretched, resulting in the gradually decreased interaction between neighboring layers, and thus the decreased stress growth rate.”

4. In the sentence “As shown in Supplementary Fig. S, in some local areas at the wrinkle, the nanowires were stretched to be more and more straight, forming local nanowire bridges”, “Supplementary Fig. S” should be “Supplementary Fig. 5”.

Response: We have changed “S” to a number, and “Supplementary Fig. 5” was revised into “Supplementary Fig. 6”, as shown below.

“As shown in Supplementary Fig. 6, in some local areas at the wrinkle, the nanowires were stretched to be more and more straight, forming local nanowire bridges.”

5. The sentence “The laminated aerogel thus represents one of the strongest resilient ceramic aerogels with strengths and modulus of up to an order of magnitude larger than previously reported ceramic aerogels under complex stress states (compressive, tensile and bending) but without compromising its flexibility and thermal insulation significantly (Fig. 4i)” with citations, should be moved to the paragraph above the conclusion part. And the related expression in the conclusion part should be revised.

Response: We have revised the related contents, as shown below.

“Benefiting from its special microstructures, the laminated aerogel thus represents one of the strongest resilient ceramic aerogels with strengths and modulus of up to an order of magnitude larger than previously reported ceramic aerogels^{14,17,30,38,39} under various deformations (including compressive, tensile and bending deformations) but without

compromising its flexibility and thermal insulation significantly (Fig. 5i).”

6. All the chemicals used in the paper, e.g. methyltrimethoxysilane and dimethyldimethoxysilane, their information (concentration, provider etc.) should be provided.

Response: We have added the information of the chemicals in the revised manuscript, as shown below.

“Thyltrimethoxysilane (MTMS, 99% purity, Meryer (Shanghai) Chemical Technology Co., Ltd., China) and dimethyldimethoxysilane (DMDMS, 99% purity, Meryer (Shanghai) Chemical Technology Co., Ltd., China,) were used as the raw materials for the synthesis of siloxane xerogel. The weight ratio of MTMS and DMDMS was 1:4. During the synthesis, ethanol ($\geq 99.7\%$ purity, Sinopharm Chemical Reagent Co., Ltd., China), deionized water and nitric acid (65.0~68.0% concentration, Sinopharm Chemical Reagent Co., Ltd., China) were used as the solvent, hydrolytic reagent and catalyst, respectively.”

Reviewer #2 (Remarks to the Author):

The authors reported a set of properties of laminated aerogel materials obtained by stacking pieces of nanowire aerogel under ethanol. The original premise that the resultant materials mimics silk cocoon structure is a significant stretch. The authors did not provide supporting data that the layers of aerogels had significant "bonding" to make the laminates any different from mere stacking the individual layers. The work in its current form lacks sufficient novelty. Some specific comments are listed below.

Response: Dear reviewer, thank you for your valuable suggestions. To address your concerns, we have carefully revised our manuscript according to your suggestions. In the revised manuscript, we used optical microscope to observe the entire formation process of the laminated aerogel. By analyzing the experimental results and other previous works, we elaborated the physics of the assembly process. According to your suggestions, through analyzing the formation mechanism, microstructure, pore size distribution, interlayer bonding, mechanical properties, deformation mechanisms, and thermal conducting behavior, we clearly show the advantages of the present laminated aerogel. We also addressed the limitations of the present work. The details are shown below.

1. The authors used the word "extreme" as several places in reference to temperature and stress level without defining for the sake of readers what this term really means. I suggest that either the authors refrain from using such a term or provide a clear definition. A similar issue is found with the term "ultralow".

Response: We have modified these expressions. "Extreme stress" and "complex stress states" were replaced by "compressive, tensile, and bending deformations". "Extreme temperature" was replaced by "-196 °C in liquid nitrogen to above 1200 °C in butane blow torch". "Ultralow" is replaced with "low" or the specific numerical values.

2. p.3, "It shows a high compressive modulus of 230 kPa, high compressive stress of 1366 kPa at 80% strain, high tensile stress of 325 kPa and modulus of 4.65 MPa, high bending strength of 260 kPa, which are several to tens of times higher than other resilient ceramic aerogel." These values do not contain measurement errors, such as standard deviation values. These should be reported.

Response: Thanks for your suggestions. We have added standard deviation values to the modulus and stress values based on three stress-strain curves for each test.

"It shows a high compressive modulus of 221.8 ± 32.7 kPa, high compressive stress of 1254.5 ± 116.3 kPa at 80% strain, high tensile stress of 398.5 ± 83.4 kPa and modulus of 4.855 ± 0.111 MPa, high bending strength of 260.5 ± 11.4 kPa, which are several to tens of times higher than other resilient ceramic aerogels¹²⁻¹⁹, showing improved load-bearing capacity under diverse deformations and making it easily to withstand the handling stress."

3. The sentence quoted in #2 above does not present the work in good stead. Are the above properties enough? A factor of "several to tens of times higher" of low numbers may be well-below what is expected. I ask the authors to list a set of representative target properties here. In my view, porous aerogel materials with 98% porosity as in this work would never be used in load bearing applications. Instead, the question should be if the aerogel products can withstand handling stress? In this context, the authors should define a range of handling stresses.

Response: We thanks the reviewer for the valuable suggestion. Besides the handling stress as you mentioned, load bearing performance is also required in some special applications such as the elastic thermal sealing materials for the hatch of a hypersonic aircraft (Ref 20 and 21). The thermal sealing of the hatch is an important safeguard to prevent the parts inside the hatch from being destroyed by the high-temperature gas flow. The position of the contact surface will inevitably bring about the reaction force of the elastic seal, and this force will also have some influence on the sealing effect. According to the data, the hatch is subjected to a force of about 1.8 N/mm during the opening and closing process. For hypersonic vehicles, due to their special operating environment and extreme conditions, they can be divided into five phases according to their functions and loads: ground preparation phase, launch phase, in-orbit segment, return segment and landing segment. The hatch state and the main load bearing for each phase are shown in Table R1, and the process requires consideration of multiple design constraints and multiple functional requirements. Therefore, the load-bearing design of thermal sealing materials is important to prevent the seal from crushing to produce permanent deformation.

Table R1. Functional profile analysis of the payload bay hatch at different stages.

Mission Phase	Main functional roles	Load bearing
Ground stage	Open with the help of auxiliary tooling	Ground operating loads and environmental conditions
Launch phase	Closed position, reliable locking	Carrier load, vibration, quasi-static load, differential pressure inside and outside the cabin
In-orbit operation phase	Hatch unlocking and unfolding, multiple spreading and closing in orbit	Space environment load (vacuum, high and low temperature, UV irradiation) variable orbit impact load
Return Phase	Closed position, reliable locking	Pneumatic power, pneumatic heat load, noise
Landing phase	Closed position, reliable locking	Landing shock load

Therefore, according to your suggestions, we have added some content to address the requirement of load-bearing capacity for the aerogel in the introduction part, as shown below.

“However, the strength and modulus of these modified aerogels are usually several

to tens of kPa^{6,7,12-19}, which are too weak to provide adequate load-bearing capacity during **handling process** and services. For example, thermal sealing materials for the hatch of a hypersonic aircraft not only requires good thermal insulation performance and elasticity but also sufficient load-bearing capacity to withstand the external aerodynamics, vibrations, and the force during opening and closing the hatch^{20,21}.”

Ref.

20. Guo Y., Chen L. & Zhou, Y. A review of the development of sealing materials and measurement and control simulation technology for typical hypersonic vehicle positions. Proceedings of the 2022 International Conference on Smart Manufacturing and Material Processing (SMMP2022), pp. 86–102, (IOS Press, 2022).

21. Dunlap Jr P. H., Steinetz B. M., Curry D. M., DeMange J. J., Rivers H. K. & Hsu S. Y. Investigations of a control surface seal for reentry vehicles. *J. Spacecr. Rockets* **40**, 570–583 (2003).

4. p.3, "The flexibility also maintains under 1200 °C butane blow torch and at temperature as low as -196 °C in liquid nitrogen." The statement/claim is not convincing without some real stress vs. strain data.

Response: Thanks for your carefully review. As you can see from Figs. 5a and b, and Supplementary Movies S6 and 7, the aerogel really exhibits reversible bending deformation under the butane blow torch and in liquid nitrogen. This kind of characterization is normally used to illustrate the thermal stability of the aerogel, as can be seen in other references¹⁴⁻¹⁶.

As you said that the stress-strain curve is missing. This is because that the tester cannot work under these conditions. Therefore, to elucidate the limitation of such characterization, we have added a note to address the shortage of the present results.

“The flexibility also maintains under 1200 °C butane blow torch and at temperature as low as -196 °C in liquid nitrogen. **However, it should be note that to further reflect the mechanical properties of the aerogel under such working temperature needs more precious characterization**”.

Ref.

14. Si Y., Wang X., Dou L., Yu J. & Ding B. Ultralight and fire-resistant ceramic nanofibrous aerogels with temperature-invariant superelasticity. *Sci. Adv.* **4**, eaas8925 (2018).

15. Li G., et al. Boron nitride aerogels with super-flexibility ranging from liquid nitrogen temperature to 1000°C. *Adv. Funct. Mater.* **29**, 1900188 (2019).

16. Dou L., et al. Interweaved cellular structured ceramic nanofibrous aerogels with superior bendability and compressibility. *Adv. Funct. Mater.* **30**, 2005928 (2020).

5. The raw SiC-SiO_x nanowire aerogel was reported in a previous publication, Ref. 28. This presents a question on the novelty of the present contribution.

Response: The aerogel reported in Ref.28 is only the raw materials used in this work. Based on your comments, we have carefully investigated the formation mechanism, and pore size distributions of the laminated aerogel. Combining with the mechanical properties, it could be seen that the present laminated aerogel is different from the raw SiC-SiO_x nanowire aerogel reported in Ref. 28 (which is updated as Ref. 30 in the revised manuscript). The details are shown below.

They are different with each other in both microstructures and properties.

1) As shown in Figure R5, the raw SiC-SiO_x nanowire aerogel in Ref.30 shows an isotropic microstructure, in which the nanowire distributed randomly in both the surface and cross section directions, and the pores assembled by nanowires is very obvious. While the present laminated aerogel is an anisotropic microstructure, in which the nanowires distributed randomly in the surface but aligned in the cross-section direction. Moreover, the nanowires in the laminated aerogel show denser packing than those in the raw aerogel. The density of the laminated aerogel is 50 mg/cm³, while that of the raw aerogel is 5.6 mg/cm³.

Figure R6. Cross section and plane view of (a, b) the raw aerogel³⁰ and (c, d) the laminated aerogel, respectively.

2) As shown in Figure R6a, four peaks at 457 nm, 2.1 μm, 36.2 μm, and 89.5 μm were observed in pore diameter distribution curve of the laminated aerogel. For comparison, in the raw SiC-SiO_x nanowire aerogel, the pore diameters were above 4 μm with a pore

diameter peak of 142.4 μm . These results indicate that in the laminated aerogel the pore size shows further decrease, which can be attributed to the denser packing of the nanowires in the laminated aerogel. According to previous study, when the pore size is below 1 μm , gas conduction is reduced dramatically because of the remarkable Knudsen effect. Therefore, gas conduction in the laminated aerogel is smaller than that in the raw nanowire aerogel.

Figure R6. Pore diameter distributions in (a) the laminated aerogel and (b) the raw aerogel, respectively. The laminated aerogel showing a decrease of the pore diameter.

3) The difference in the microstructure and pore size distribution is attributed to the formation mechanism of the laminated aerogel. The mechanism is illustrated in detail in the revised manuscript and the response to **Comment 6** and the revised manuscript. During the formation process, the raw aerogel showed severe vertical shrinkage and the nanowires show radial reorientation under the action of the ethanol evaporation-generated radial capillary flow.

4) Benefiting from the special microstructure, the present laminated aerogel thus represents one of the strongest resilient ceramic aerogels with strengths and modulus of up to an order of magnitude larger than previously reported ceramic aerogels under various deformations (including compressive, tensile and bending deformations) but without compromising its flexibility and thermal insulation significantly (Figure R7).

Figure R7. Comparison of the thermal and mechanical properties of the laminated aerogel and other highly stretchable, bendable and compressible ceramic aerogels^{14,17,30,38,39}, indicating a similar thermal property but much higher strength under tension, bending and compression.

Altogether, by analyzing the microstructure, pore size distribution (related to thermal conductivity), formation mechanism, and mechanical properties, the present laminated aerogel shows its own characters, verifying the novelty of the present work.

Ref.

14. Si Y., Wang X., Dou L., Yu J. & Ding B. Ultralight and fire-resistant ceramic nanofibrous aerogels with temperature-invariant superelasticity. *Sci. Adv.* **4**, eaas8925 (2018).
17. Guo J., et al. Hypocrystalline ceramic aerogels for thermal insulation at extreme conditions. *Nature* **606**, 909–916 (2022).
25. Su L., et al. Resilient Si₃N₄ nanobelt aerogel as fire-resistant and electromagnetic wave-transparent thermal insulator. *ACS Appl. Mater. Interfaces* **11**, 15795–15803 (2019).
30. Su L., et al. Highly stretchable, crack-insensitive and compressible ceramic aerogel. *ACS Nano* **15**, 18354–18362 (2021).
39. Cheng X., Liu Y.-T., Si Y., Yu J. & Ding B. Direct synthesis of highly stretchable ceramic nanofibrous aerogels via 3D reaction electrospinning. *Nat. Commun.* **13**, 2637 (2022).

6. The process of fabrication of laminated aerogel is too simplistic. Ethanol evaporation from the pores brings out "self-assembly" of the particles through capillarity. It is a common knowledge that ethanol evaporation leads to densification of the aerogel. Authors needs to elaborate the physics of assembly process.

Response: To demonstrate clearly the physics of the assembly process, we observed the assembly process under an optical microscope. The formation mechanism is illustrated clearly through analyzing the experimental observations according previous theories reported in other references. We have rewritten the formation process of the laminated aerogel in the revised manuscript. The revised Figure 1 and content are shown below.

We used a homemade highly compressible and stretchable SiC-SiO_x nanowire aerogel paper³⁰ with a density of 5.6 mg/cm³ as the raw materials to prepare the laminated SiC-SiO_x nanowire aerogel. The raw SiC-SiO_x nanowire aerogel paper was prepared through a chemical vapor deposition method, which is illustrated in detail in Methods and our previous work³⁰. To realize the construction of the laminated microstructure, a facile capillary force-induced self-assembly method was developed. Fig. 1a illustrates the fabrication process of the laminated ceramic aerogel. Firstly, the raw aerogel paper was cut into pieces and then they were stacked layer by layer to form a bulk aerogel. The bulk aerogel was immersed in ethanol. After it was fully infiltrated, the aerogel was put out and then dried naturally. During the evaporation of ethanol in the drying process, the self-assembly of the nanowires induced by the capillary force took place, resulting in the formation of the laminated SiC-SiO_x nanowire aerogel. Fig. 1b shows the macroscopic morphology of the as-prepared laminated aerogel standing on the surface of a leaf, **in which we can observe the laminated structure of the aerogel**

obviously.

To insights into the formation mechanism of the laminated structure, we attached a piece of the raw SiC-SiO_x nanowire aerogel on a solid substrate (glass slide) mounted on a heating element with a temperature of 50 °C (Figs. 1c and d). Then we dropped ethanol on the aerogel paper and observed the microstructure evolution from the cross section during the evaporation of ethanol. As shown in Fig. 1e and Supplementary Movie 1, during the drying process, a piece of aerogel scrap flows along the radial direction. This phenomenon indicates that the ethanol flows along the radial direction of the aerogel paper, resulting in a radial capillary flow. This is because that the liquid flow is more likely to move towards the solvent-substrate-air line³¹⁻³⁴. Under the action of the radial capillary flow, the nanowires were gradually aligned along the direction of the liquid flow (Fig. 1d), which could be confirmed by the transverse wrinkles appeared in the cross section during the evaporation of the ethanol (Fig. 1f and Supplementary Movie 2). This kind of capillary flow-induced reorientation behavior was also observed in a tungsten oxide nanowire system during drying³⁵. Simultaneously, with the evaporation of the ethanol, severely shrinkage of the aerogel in the vertical direction, resulting in the densification and further reorientation of the aerogel (Figs. 1d and f). The shrinkage of the aerogel is to compensate the ethanol loss caused by the evaporation at the solvent-substrate-air line³⁵. Therefore, it could be concluded that the radial capillary flow and vertical shrinkage of the aerogel result in the reorientation of the nanowires along the radial direction and thus the formation of the laminated microstructure.

Figure R8. (Fig. 1 in revised manuscript). Preparation process, formation mechanism and macroscopic morphology of the laminated SiC-SiO_x nanowire aerogel. a, Schematic illustration of the preparation process. **b**, Macroscopic morphology of the aerogel. **c, d**, schematic illustration of the formation mechanism of the laminated aerogel during the during process. The radial capillary flow and vertical shrinkage of the aerogel working together to result in the formation of the laminated aerogel. **e**, Observation of the radial ethanol flow during the evaporation by using an optical microscope. **f**, Optical microscopy images showing the vertical volume shrinkage and reorientation of the nanowires during evaporation.

Ref.

31. Huang J., Fan R., Connor S. & Yang P. One-step patterning of aligned nanowire arrays by programmed dip coating. *Angew. Chem. Int. Ed.* **46**, 2414–2417 (2007).
32. Deegan R. D., Bakajin O., Dupont T. F., Huber G., Nagel S. R. & Witten T. A. Capillary flow as the cause of ring stains from dried liquid drops. *Nature* **389**, 827–829 (1997).
33. Deegan R. D. Pattern formation in drying drops. *Phys. Rev. E.* **61**, 475–485 (2000).
34. Deegan R. D., Bakajin O., Dupont T. F., Huber G., Nagel S. R. & Witten T. A. Contact line deposits in an evaporating drop. *Phys. Rev. E* **62**, 756–765 (2000).
35. Cheng W. & Niederberger, M. Evaporation-induced self-assembly of ultrathin tungsten oxide nanowires over a large scale for ultraviolet photodetector. *Langmuir* **32**, 2474–2481 (2016).

7. The difference of images in Figure 1(d) and (f) is not clear - the former uses a scale bar of 2 micrometer, while the latter with 1 micrometer. Yet, the figure in 1(f) is a lot more revealing and does not show broken nanowires, as in 1(d). Why?

Response: Figs. 1 c and d are the cross-section SEM images, while Fig. 1e and f are the plane-view SEM images. Fig. 1d is the amplified image of the area marked in 1c, Fig. 1f is the amplified image of the area marked in 1e.

The sample used to observe the cross-section microstructure was the fracture surface of the laminated aerogel. Therefore, the observed broken nanowires were the fractured and pulling-out nanowires, as that observed during the fracture process (Figure R10). While the plane view SEM images were observed from the surface of the aerogel perpendicular to the nanowire aligned layers. Because the nanowire so long that we cannot observe the root and tip of the a nanowire, Figure 1f does not shown broken nanowires as that in Figure 1d.

In the revised manuscript, Figs. 1c to f were revised into Figs. 2a to d, as shown below.

Figure R9. (Fig. 1c to f in the previous version of the manuscript, and Fig. 2a to d in the revised manuscript). **c**, Cross-section morphology of the aerogel showing the laminated structure. **d**, Amplified cross-section SEM image of the marked area in **(c)** showing the oriented SiC-SiO_x nanowires in each layer. **e**, Plane view of the aerogel showing the wrinkled structures in each nanowire layer. **f**, Amplified SEM images of the marked area in **(e)** showing the well-interconnected SiC-SiO_x nanowires and bundles assembled nanowire layer.

Figure R10. (Figure 4 e and f in the revised manuscript) **a**. Crack deflection taking place at the wrinkles during fracture. **b**. The amplified microstructure in the inset in **(a)** shows the pulling-out and nanowire bridging behavior during the crack propagation.

8. p.5, "As shown in Fig. 2a, the aerogel can be compressed to 20%, 40% and 60% strain consecutively, and then recovers to almost its original size, demonstrating the good reversible compressibility." This claim cannot be true. Figure 2(a) shows hysteresis, although authors did not clearly indicate compression part on the figure. Thus, these materials underwent significant structural damage. The inset in this figure without a scale bar cannot clearly support the claim. I ask authors to present stress-strain diagram at low strain (less than 10%). Also, how much time does the shape recovery take?

Response: Thanks for your suggestions. We provided the compressive stress-strain diagrams at low strain. As shown in Figure a, there is no permanent deformation before 40% strain, and the permanent deformation at set strain of 60% and 80% are about 6%. Figure b shows the stress-strain curve of the aerogel during 100 cyclic fatigue test at set strain of 40%. During the first 10 cycles, there is rarely permanent deformation. When the aerogel was compressed for 20 cycles, the permanent strain is about 6%. When the compression cycle reaches 100, a permanent strain of about 10% is achieved. These figures were added in the revised Supplementary Information as Supplementary Fig. 2.

Figure R11. (Supplementary Fig. 2 in the revised Supplementary Information) | Stress-strain diagrams at low strains. **a**, consecutive compression at set strains of 20%, 40%, 60% and 80%. **b**, 100 cyclic fatigue compressive tests at set strain of 40%.

The revised content in the manuscript.

“As shown in Fig. 3a and Supplementary Fig. 2a, the aerogel can be compressed to 20% and 40% strain consecutively, and then recovers to almost its original size, demonstrating the good reversible compressibility. When the compressive strain reaches 60% and 80%, only a small permanent deformation of ~ 6% is observed.”

“Fig. 3c and Supplementary Fig. 2b show the stress-strain curve of the aerogel during 100 cyclic fatigue tests at set strain of 40%. During the first 10 cycles, there is rarely permanent deformation. When the aerogel was compressed for 20 cycles, the permanent strain is about 6%. Even the compression cycle reaches 100, the permanent strain is only ~10%.”

9. What is missing from this work is a test to evaluate the interlayer "adhesion" that the authors claimed to have achieved. In my view, the properties that the authors reported in this work would be shown by individual layers of the paper.

Response: We characterized the interlayer adhesion through testing a piece of the laminated aerogel by applying tensile stress perpendicular to the nanowire layers. The result is shown in Figure R12. The maximum adhesion stress is about 2.25 kPa. Even the aerogel was delaminated under the test, the local area of the aerogel was connected with each other. This characterization was added in the revised manuscript. Figure R12 is added as Fig. 2i.

Figure R12 (Fig. 2i in the revised manuscript). The tensile stress-displacement curve of

the aerogel along the direction perpendicular to the nanowire layers, showing the adhesion stress between neighboring layers. Insets showing the macroscopic morphology of the aerogel during the test.

Added content in the revised manuscript:

“Worth noting that the neighboring nanowire layers are not simply stacked. We characterized the interlayer adhesion through testing a piece of the aerogel by applying tensile stress perpendicular to the nanowire layers. The result in Fig. 2i show that the maximum adhesion stress is about 2.25 kPa. Even the aerogel was delaminated during the test, some local parts between the neighboring layers was connected with each other.”

For comparison, the tensile strength along the nanowire aligned direction is 398.5 ± 83.4 kPa. This kind of anisotropic mechanical properties is very similar to the mechanical properties of natural silk cocoon. According to the reference (Acta Biomaterialia 8 (2012) 2620–2627), the average peel off load between the layers in silk cocoon is only 0.32N, while the strength of the silk cocoon layers is 14.6 to 59.7 MPa. Although the very low peel off load, previous work show that both the interlayer bonding and the mechanical property of individual layer would influence the mechanical properties of the cocoon (Acta Biomaterialia 8 (2012) 2620–2627). Therefore, the properties of the laminated aerogel are determined by both the mechanical properties of the nanowire layers and the interlayer bonding.

If the interlayer adhesion is enhanced, the flexibility of the laminated aerogel cannot maintain. For example (Figure R2), in a laminated SiC@SiO₂ nanowire aerogel prepared by hot-pressing method, the interlayers bonding is largely enhanced, but the structure showed brittle fracture under bending deformation. This result further shows the advantage of the present microstructure design.

Ref: Fujia Chen, David Porter, Fritz Vollrath. Silk cocoon (*Bombyx mori*): Multi-layer structure and mechanical properties. Acta Biomaterialia 8 (2012) 2620–2627.

10. I do not see any explanation of heat transport in these materials. No new information is presented. Readers would love to relate heat transport to pore size and its distribution.

Response: We measured the pore distribution of both the laminated aerogel and raw aerogel with randomly distributed nanowires. Compared with the raw nanowire aerogel, the pore size is decreased in the laminated aerogel, even some nanosized pores appeared. We discussed the effect of pore distribution on the thermal insulation performance of the aerogel. The related contents were revised and the details is shown below.

Revised content:

The measured result shows that the laminated SiC nanowire aerogel exhibit a low room-temperature thermal conductivity of about 39.3 mW/m·K in the direction perpendicular to the laminated structure (Supplementary Table 1), which is comparable to that of the silk cocoons^{27,28}, showing good thermal insulation performance. As is

known, solid conduction, gas conduction, gas convection, radiation are the factors that influence the thermal conductivity in porous materials. The contribution of radiation is negligible at room temperature. Gas convection is also inhibited in porous material with pore size less than 4 mm^{40,41}. Measured result (Fig. 2j) shows that the pore diameter is less than 400 μm in the laminated aerogel, indicating there is rarely gas convection. Therefore, thermal conductivity of the laminated aerogel can be attributed to the solid conduction and gas conduction.

Figure R14. (Fig. 2j and k in the revised manuscript). Pore diameter distributions in (a) the laminated aerogel and (b) the raw aerogel, respectively. The laminated aerogel showing a decrease of the pore diameter.

Solid conduction of in the laminated SiC-SiO_x nanowire aerogel depends on the phonon conduction. The high porosity (~98%) can block and the nanowires-assembled tortuous structure can prolong the phonon conducting pathway, thus reducing solid conduction. Moreover, the phonon conduction barriers caused by the amorphous SiO_x sub-nanowire and the stacking faults in the SiC sub-nanowire can further reduce solid conduction^{13,30,42–45}. These factors are responsible for the decrease of thermal conductivity. For comparison, the density of the laminated aerogel (50 mg/cm³) is about 10 times higher than that of the raw SiC-SiO_x nanowire aerogels with random microstructure (with a density of 5.6 mg/cm³ and a thermal conductivity of 28.4 mW/m·K³⁰), which indicates that the solid conduction in the laminated aerogel might be 10 times higher than that of the raw SiC-SiO_x nanowire aerogels. However, the thermal conductivity only shows an increase factor of ~1.37. Therefore, there are other factors that are responsible for the good thermal insulation performance.

Compared with the pore diameter distribution in the raw SiC-SiO_x nanowire aerogel (mainly in the range of 4 to 400 μm, Fig. 2k), in the laminated aerogel nanosized pores distributed between 100 nm to 1 μm appears (Fig. 2j). According to previous study, when the pore size is below 1 μm, gas conduction is reduced dramatically because of the remarkable Knudsen effect⁴¹. Therefore, gas conduction in the laminated aerogel is smaller than that in the raw nanowire aerogel. However, because of the limited volume ratio (only about 2%) of nanosized pores in the laminated aerogel, the reduced gas conduction itself cannot explain the good thermal insulation performance of the present laminated aerogel.

We then observed the heat transportation behavior in the laminated aerogel. For comparison, we also recorded the heat distribution in an isotropic SiC-SiO_x nanowire aerogel. As shown in Supplementary Fig.7, a cylinder heater with a diameter of 2 mm was used as the heat source to generate heat on the surface of the aerogel samples. It could be seen that during the heating process the 25 °C isothermal line shows an anisotropic shape in the laminated aerogel (Figs. 5d and f, Supplementary Fig. 8 and Supplementary Movie 8), which is obviously different from the isotropic shape in the isotropic aerogel (Figs. 5e and f, Supplementary Fig. 9 and Supplementary Movie 9). We then calculated the anisotropic factor of the 25 °C isothermal lines (The ratio between the horizontal length and vertical length in the shape of the 25 °C isothermal lines). As shown in Fig. 5g and Supplementary Table 2 and 3, with the gradual increase of the heat source temperature during the heating process, the anisotropic factor of the laminated aerogel gradually stabilizes at around 1.3, while that of the isotropic aerogel is around 1.0 (Fig. 5g). These observations indicate the anisotropic thermal conducting behavior of the laminated aerogel. Such anisotropic heat flowing behavior could increase the heat transportation along the nanowire layers, thus reducing the heat transportation and increasing the thermal insulation in the direction perpendicular to the laminated structure (Fig. 5h). **Therefore, the good thermal insulation of the laminated aerogel is attributed to not only the high porosity, and the phonon conducting barriers but also the existence of nanosized pores and the laminated microstructure induced anisotropic heat transport behavior.**

Ref:

40. Collishaw P. & Evans J. An assessment of expressions for the apparent thermal conductivity of cellular materials. *J. Mater. Sci.* **29**, 2261–2273 (1994).
41. Liu H. & Zhao X. Thermal conductivity analysis of high porosity structures with open and closed pores. *Int. J. Heat. Mass. Tran.* **183**, 122089 (2022).
42. Yan X., et al. Single-defect phonons imaged by electron microscopy. *Nature* **589**, 65–69 (2021).
43. Su L., et al. Anisotropic and hierarchical SiC@SiO₂ nanowire aerogel with exceptional stiffness and stability for thermal superinsulation. *Sci. Adv.* **6**, eaay6689 (2020).
44. Donadio D. & Galli G. Temperature dependence of the thermal conductivity of thin silicon nanowires. *Nano Lett.* **10**, 847–851 (2010).
45. Valentín L. A., et al. A comprehensive study of thermo-electric and transport properties of β -silicon carbide nanowires. *J. Appl. Phys.* **114**, 184301 (2013).

Reviewer #1 (Remarks to the Author):

I appreciate the author's efforts to improve the paper according to the comments, I still have a comment on the authors' response.

In Figure 2, the authors added the pore size distribution measured by the mercury intrusion porosimetry, as you know this method is normally used for relatively rigid samples, the deformation during the intrusion can largely influence the results. The laminated gels are very elastic, I doubt the results from the mercury test, it would be good if gas sorption (BET, N₂ and Krypton) could be carried out for such comparison, and for the aerogel materials, it will also be good to know the gas sorption behavior to know the specific surface area and pore size.

Reviewer #3 (Remarks to the Author):

The authors have addressed all comments from reviewer #2. However, the main concern about the "The authors did not provide supporting data that the layers of aerogels had significant "bonding" to make the laminates any different from mere stacking the individual layers". Authors have explained the detailed experimental procedures of laminated aerogel synthesis by dispersing aerogel paper in ethanol. It is still not sure that how there two SiC-SiO₂ aerogel papers are connected or bonding each other to make a strong interaction. It would be recommended to explain how it is connected like by physically or chemically and how these structures are related to its mechanical properties. Also, it would be important to emphasize the novelty of the work, as the authors have mentioned that the nanowire (SiC-SiO_x) aerogel paper, from author's previous work (Ref [30] ACS Nano 2021, 15, 11, 18354–18362), was used to make laminated nanowire aerogel.

Point-by-Point Response to the Reviewers' Comments

Title: Strong yet Flexible Ceramic Aerogel

Manuscript ID: NCOMMS-22-48377A

COMMENTS TO AUTHOR:

Reviewer #1 (Remarks to the Author):

I appreciate the author's efforts to improve the paper according to the comments, I still have a comment on the authors' response. In Figure 2, the authors added the pore size distribution measured by the mercury intrusion porosimetry, as you know this method is normally used for relatively rigid samples, the deformation during the intrusion can largely influence the results. The laminated gels are very elastic, I doubt the results from the mercury test, it would be good if gas sorption (BET, N₂ and Krypton) could be carried out for such comparison, and for the aerogel materials, it will also be good to know the gas sorption behavior to know the specific surface area and pore size.

Response: Thanks for your appreciation to our revision. According to your comments, we have added the N₂ sorption results in the revised manuscript, and also added discussion on the results measured by mercury intrusion porosimetry.

Figures R1a and b show the N₂ sorption isotherms at 77 K and pore size distribution derived from Barrett–Joyner–Halenda (BJH) analysis, respectively. The calculated specific surface areas of the laminated aerogel and raw aerogel are 15.4 and 29.1 m²/g, respectively. And the average pore sizes are 11.8 and 14.3 nm, respectively. Both the specific surface area and pore size of the laminated aerogel show decrease when compared with those of the raw aerogel. The decrease of the specific surface area is related to the severe volume shrinkage of the raw aerogel during the preparation process, which results from the more compact interconnection between neighboring nanowires (as shown in Figure R2).

Figure R1. (a) N₂ sorption isotherms at 77 K and (b) pore size distribution derived from Barrett–Joyner–Halenda (BJH) analysis, respectively

Figure R2. Cross section and plane view of (a, b) the raw aerogel and (c, d) the laminated aerogel, respectively. The laminated aerogel shows a more compact structure.

As the reviewer pointed that the elastic deformation during the intrusion of mercury during the test could influence the results. From the compressive property of the laminated aerogel, we can see that the maximum stress can reach an average value of 1254.5 ± 116.3 kPa at 80% strain. It is sure that before the pressure of mercury reaches more than 1250 kPa the structure of the laminated aerogel is integral, except for the elastic deformation.

Herein, we assume that the pore shape is spherical, then the volume shrinkage ratio, X , during the mercury intrusion process can be calculated by equation 1.

$$X = \frac{\frac{4}{3}\pi R^3 - \frac{4}{3}\pi r^3}{\frac{4}{3}\pi R^3} \quad (1)$$

Where R is the real pore radius of the pores, r is the tested pore radius. The real pore radius be calculated by equation 2.

$$R = \left(\frac{1}{1-X}\right)^{\frac{1}{3}} r \quad (2)$$

Because of the nearly zero Poisson ratio, the volume shrinkage at $x\%$ compressive strain is about $x\%$. If there is no volume shrinkage, $R=r$. If the volume shrinkage is 60%, then $R=1.35r$; which indicates that the real pore size is 1.35 times that of tested one. If the volume shrinkage is 80%, then $R=1.71r$.

Figure R3 shows the relationship between the tested pore size and the applied pressure to the mercury (in the range of 0 to 1250 kPa). It could be seen that the maximum tested pore size is $355 \mu\text{m}$ (the pressure of mercury is 3.5 kPa) and the minimum tested pore size is $1 \mu\text{m}$ (the pressure of mercury is 1250 kPa).

Figure R3. The relationship between the tested pore size and the applied pressure to the mercury (in the range of 0 to 1250 kPa).

Considering the elastic deformation of the aerogel during the test and the mercury intrusion porosimetry results displayed in Figure R5, it could be concluded that the pore size in the laminated aerogel is below 355 μm .

Figure R4. The tested pore size distribution in the laminated aerogel by mercury intrusion porosimetry.

Figure R5 shows the relationship between the tested pore size (in the range of 0 to 100 nm) and the applied pressure to the mercury. It could be seen that if one wants to measure the pore size below 100 nm, the pressure is larger than 10000 kPa. At such high pressure, the aerogel might be fractured. Therefore, mercury intrusion porosimetry is not suitable for measure the nanosized pore in the present laminated aerogel.

Figure R5. The relationship between the tested pore size (in the range of 0 to 100 nm) and the applied pressure to the mercury.

Based on the above analysis, we have revised the related part and added Supplementary Discussion in the Supplementary Information, as shown below.

Revised content in the main manuscript:

Fig. 2j and k show the N₂ sorption isotherms at 77 K and pore size distribution derived from Barrett–Joyner–Halenda (BJH) analysis, respectively. The calculated specific surface areas of the laminated aerogel and raw aerogel are 15.4 and 29.1 m²/g, respectively. And the average pore sizes are 11.8 and 14.3 nm, respectively. Both the specific surface area and pore size of the laminated aerogel show decrease when compared with those of the raw aerogel. The decrease of the specific surface area is related to the severe volume shrinkage of the raw aerogel during the preparation process, which results from the more compact interconnection between neighboring nanowires. We also used mercury intrusion porosimetry to investigate the macropores in the laminated aerogel, the results show that the pores width is below 355 μm (a detail discussion is presented in Supplementary Discussion).

Revised Fig. 2| j. N₂ sorption isotherms at 77 K and **k.** pore size distribution derived from Barrett–Joyner–Halenda (BJH) analysis, respectively

Added Supplementary Information.

Supplementary Discussion.

From the compressive property of the laminated aerogel, we can see that the maximum stress can reach an average value of 1254.5±116.3 kPa at 80% strain. It is sure that before the pressure of mercury reaches more than 1250 kPa the structure of the laminated aerogel is integral, except for the elastic deformation.

Herein, we assume that the pore shape is spherical, then the volume shrinkage, X, during the mercury intrusion process can be calculated by equation 1.

$$X = \frac{\frac{4}{3}\pi R^3 - \frac{4}{3}\pi r^3}{\frac{4}{3}\pi R^3} \quad (1)$$

Where R is the real pore radius of the pores, r is the tested pore radius. The real pore radius be calculated by equation 2.

$$R = \left(\frac{1}{1-X} \right)^{\frac{1}{3}} r \quad (2)$$

Because of the nearly zero Poisson ratio, the volume shrinkage at $x\%$ compressive strain is about $x\%$. If there is no volume shrinkage, $R=r$. If the volume shrinkage is 60%, then $R=1.35r$, which indicates that the real pore size is 1.35 times that of tested one. If the volume shrinkage is 80%, then then $R=1.71r$.

Supplementary Fig. 13 shows the relationship between the tested pore size and the applied pressure to the mercury (in the range of 0 to 1250 kPa). It could be seen that the maximum tested pore size is 355 μm (the pressure of mercury is 3.5 kPa) and the minimum tested pore size is 1 μm (the pressure of mercury is 1250 kPa).

Considering the elastic deformation of the aerogel during the test and the mercury intrusion porosimetry results displayed in Supplementary Fig. 14, it could be concluded that the pore size in the laminated aerogel is below 355 μm .

Supplementary Fig. 13 | The relationship between the tested pore size and the applied pressure to the mercury (in the range of 0 to 1250 kPa).

Supplementary Fig. 14 | The tested pore size distribution in the laminated aerogel by mercury intrusion porosimetry.

Reviewer #3 (Remarks to the Author):

The authors have addressed all comments from reviewer #2. However, the main concern about the “The authors did not provide supporting data that the layers of aerogels had significant "bonding" to make the laminates any different from mere stacking the individual layers”. Authors have explained the detailed experimental procedures of laminated aerogel synthesis by dispersing aerogel paper in ethanol. It is still not sure that how there two SiC-SiO₂ aerogel papers are connected or bonding each other to make a strong interaction. It would be recommended to explain how it is connected like by physically or chemically and how these structures are related to its mechanical properties. Also, it would be important to emphasize the novelty of the work, as the authors have mentioned that the nanowire (SiC-SiO_x) aerogel paper, from author’s previous work (Ref [30] ACS Nano 2021, 15, 11, 18354–18362), was used to make laminated nanowire aerogel.

Response: Thank you for your approval of our revision. We have added further characterization to illustrate the interlayer connection.

(1) The connections between neighboring layers.

The laminates are connected with each other by physically instead of mere stacking of the individual layers. As shown in Figure R6, the tested interlayer adhesive stress is about 2.25 kPa. When the aerogel was delaminated during the test, although the appearance of obvious gap between neighboring layers, local parts at the interface are still connected with each other at large applied displacement (insets in Figure R6a). This indicates the good connection between neighboring layers.

To insights into the microscale interconnection between neighboring layers, we observed the interface during the delamination process. As shown in Figure R6b, during the delamination process, bridging behaviors of the nanowires were seen clearly at the interface between the neighboring laminates. Such good bonding is formed during the sample preparation process. The large capillary force generated by the solvent evaporation can bring the connecting nanowires of different laminates together to form physical connections.

Figure R6. a. Tensile stress-displacement curve of the aerogel along the direction perpendicular to the nanowire layers, showing the adhesion stress between neighboring layers. Insets showing the macroscopic morphology of the aerogel during the test. b. Nanowire bridging behavior at the interface between neighboring layers during delamination.

(2) How these structures are related to its mechanical properties?

Although the bonding stress is low, it is worth noting that this kind of physical connection is **beneficial for the laminated to achieve flexibility under bending and tensile deformation**. The low interlayer adhesive stress could provide the nanowires greater degree of freedom to deform under tensile stress and thus help the laminated aerogel to maintain their flexibility under tensile and bending deformations (Figure R7).

Figure R7. Large strain tensile and bending deformation of the present laminated aerogel.

If the interlayer adhesion is enhanced, the flexibility of the nanowire during tensile or bending deformation is largely decreased. For example (Figure R8), in a laminated SiC@SiO₂ nanowire aerogel prepared by hot-pressing method, the interlayers bonding is largely enhanced, but the structure showed brittle fracture under bending deformation.

Figure R8. Microstructure of a nanowire-assembled laminated aerogel prepared by hot-pressing method. The nanowire is well bonded, exhibiting a strong interlayer bonding. However, the strong bonding will decrease the flexibility of the nanowire, thus resulting in the brittle bending fracture.

From the microscale, it could be seen that the present laminated aerogel can be easily rolled up, while the laminated aerogel with strong interlayer bond is easily broken by applying bending deformation (Figure R9). As shown in Figure R9a, in the rolled laminated aerogel, we can also see clearly the weaving of the laminates, which further verifies that the physical interlayer bonding can enable the nanowires good flexibility under deformation.

Figure R9. a. The microstructure morphology of the present laminated aerogel under bending. b. The laminated aerogel with strong chemical bonding is easily broken under bending.

(3) The novelty of the present work.

a) **The differences in the microstructures between the present laminated aerogel and the raw aerogel reported in Ref.30.** As shown in Figure R10, the raw SiC-SiO_x nanowire aerogel in Ref.30 shows an isotropic microstructure, in which the nanowire distributed randomly in both the surface and cross section directions, and the pores assembled by nanowires is very obvious. While the present laminated aerogel is an anisotropic microstructure, in which the nanowires distributed randomly in the surface but aligned in the cross-section direction. The nanowires in the laminated aerogel show denser packing than those in the raw aerogel. And the density of the laminated aerogel is 50 mg/cm³, while that of the raw aerogel is 5.6 mg/cm³. Therefore, they are different materials.

Figure R10. Cross section and plane view of (a, b) the raw aerogel³⁰ and (c, d) the laminated aerogel, respectively.

b) **The highest strength and modulus reported for resilient ceramic aerogel.** Benefiting from the special microstructure, the present laminated aerogel shows a high compressive modulus of 221.8±32.7 kPa, high compressive stress of 1254.5±116.3 kPa at 80% strain, high tensile stress of 398.5±83.4 kPa and modulus of 4.855±0.111 MPa, high bending strength of 260.5±11.4 kPa, which are several to tens of times higher than previous reported resilient ceramic aerogels^{12-19,30}, including the raw aerogel (Ref.30)

that is used for the preparation of the present laminated aerogel.

c) **Good flexibility under tensile, bending, buckling and compressive stresses.** Besides the large strengths and modulus, benefiting from the designed laminated microstructure with physical interlayer connection, the present aerogel also exhibits flexibility under tensile, bending, buckling and compressive deformations. It shows a ductile tensile behavior with a fracture strain of up to 20%, reversible bending deformation, resilient buckling deformation up to 80% strain, and recoverable compressibility up to 80% strain.

d) **Good thermal insulation performance.** The laminated microstructure also results in an anisotropic thermal conducting behavior, which helps it reduce heat flux in the direction perpendicular to the nanowires layers and thus gives it low thermal conductivity.

For comparison, most of the highly compressible ceramic aerogels usually show brittleness under tensile stress, while those ceramic aerogels with reversible tensile strain usually exhibits low tensile strength and modulus, and those ceramic aerogels with increased modulus and strength usually show decreased flexibility and thermal insulation.

The present laminated aerogel thus represents one of the strongest resilient ceramic aerogels with strengths and modulus of **up to an order of magnitude larger** than previously reported ceramic aerogels under various deformations (including compressive, tensile and bending deformations) but **without compromising its flexibility and thermal insulation significantly** (Figure R11). The attractive properties are achieved through the designed nanowires-assembled laminated with physical interlayer bonding. Such microstructure design enables the nanowires sufficient deformation resistance and deformability under various stress states as well as efficient thermal conducting barriers. The design strategy in the present work provides a model for resolving the conflicts among strength, deformability and thermal insulation in ceramic aerogels

Figure R11. Comparison of the thermal and mechanical properties of the laminated aerogel and other highly stretchable, bendable and compressible ceramic aerogels^{14,17,30,38,39}, indicating a similar thermal property but much higher strength under tension, bending and compression.

Ref.

12. Wang H., et al. Ultralight, scalable, and high-temperature-resilient ceramic nanofiber sponges. *Sci. Adv.* **3**, e1603170 (2017).

13. Su L., et al. Ultralight, recoverable, and high-temperature-resistant SiC nanowire aerogel. *ACS Nano* **12**, 3103–3111 (2018).
14. Si Y., Wang X., Dou L., Yu J. & Ding B. Ultralight and fire-resistant ceramic nanofibrous aerogels with temperature-invariant superelasticity. *Sci. Adv.* **4**, eaas8925 (2018).
15. Li G., et al. Boron nitride aerogels with super-flexibility ranging from liquid nitrogen temperature to 1000°C. *Adv. Funct. Mater.* **29**, 1900188 (2019).
16. Dou L., et al. Interweaved cellular structured ceramic nanofibrous aerogels with superior bendability and compressibility. *Adv. Funct. Mater.* **30**, 2005928 (2020).
17. Guo J., et al. Hypocrystalline ceramic aerogels for thermal insulation at extreme conditions. *Nature* **606**, 909–916 (2022).
18. Li L., et al. Nanograin-glass dual-phasic, elasto-flexible, fatigue-tolerant, and heat-insulating ceramic sponges at large scales. *Mater. Today* **54**, 72–82 (2022).
19. Guo P., et al. Additive manufacturing of resilient SiC nanowire aerogels. *ACS Nano* **16**, 6625–6633 (2022).
30. Su L., et al. Highly stretchable, crack-insensitive and compressible ceramic aerogel. *ACS Nano* **15**, 18354–18362 (2021).
38. Li M., et al. Stretchable and compressible Si₃N₄ nanofiber sponge with aligned microstructure for highly efficient particulate matter filtration under high-velocity airflow. *Small* **17**, 2100556 (2021).
39. Cheng X., Liu Y.-T., Si Y., Yu J. & Ding B. Direct synthesis of highly stretchable ceramic nanofibrous aerogels via 3D reaction electrospinning. *Nat. Commun.* **13**, 2637 (2022).

Based on the above analysis, we have revised the related part, as shown below.

Worth noting that the neighboring nanowire layers are bonded with each other by physically. We characterized the interlayer adhesion through testing a piece of the aerogel by applying tensile stress perpendicular to the nanowire layers. The result in Fig. 2i show that the maximum adhesion stress is about 2.25 kPa. Even the aerogel was delaminated during the test, some local parts between the neighboring layers was connected with each other. To insights into the microscale interconnection between neighboring layers, we observed the interface during the delamination process. As shown in Supplementary Fig. 2, during the delamination process, bridging behaviors of the nanowires were seen clearly at the interface between the neighboring laminates. Such good bonding is formed during the sample preparation process. The large capillary force generated by the solvent evaporation can bring the connecting nanowires of different laminates together to form physical connections. This kind of interlayer bonding could give the aerogel sufficient deformability under tensile and bending stresses, which will be discussed later.

Revised Fig. 2 | i. Tensile stress-displacement curve of the aerogel along the direction perpendicular to the nanowire layers, showing the adhesion stress between neighboring layers. Insets showing the macroscopic morphology of the aerogel during the test.

Supplementary Fig. 2 | SEM image showing the nanowire bridging behavior at the interface between neighboring layers during delamination.

...Moreover, the physical connections between neighboring layers enables the nanowires and nanowire layers good flexibility under bending deformation (as verified by the weaving behavior of the nanowire layers in the bent sample in Supplementary Fig. 8), thus resulting in the good flexibility under bending and buckling....

Supplementary Fig. 8 | The microstructure morphology of the present laminated aerogel under bending. The nanowire layers showed obvious weaving behavior under bending, indicating its flexibility.

Reviewer #1 (Remarks to the Author):

Thank you for the authors' response to the comments. Based on the N₂ sorption results, it is evident that the surface area is quite low. The low N₂ sorption data confirms the low mesoporosity of the ceramic aerogel materials. Given that the N₂ molecule can exclusively detect the mesopore structure, the average pore sizes (11.8 and 14.3 nm) calculated using the BJH method may not encompass the entire range of pore sizes. To gain a comprehensive understanding of the overall pore volume and size of the samples, I suggest combining this information with bulk density (Reichenauer, G. (2011). Structural characterization of aerogels. Aerogels handbook, 449-498.):

$$1/\text{bulk density} = V_{\text{total}} = V_{\text{micro\&meso}} + 1/\text{skeletal density}$$
$$D_{\text{average}} = 4 * V_{\text{total}} / \text{SBET}$$

It would be beneficial if the authors could integrate the N₂ sorption, mercury intrusion porosimetry, and the total pore calculation outlined above to offer insights into potential pore size distributions. This approach goes beyond merely listing two methods and discussing their results in isolation.

Reviewer #2 (Remarks to the Author):

I read this contribution with interest. I also went through reviewer comments and the author responses. I believe there is something unique in these materials. The method of aerogel fabrication by putting together thin nanowire aerogel paper is deceptively simple. This is where readers would ask why such materials show the properties that are reported in this paper. The authors indicated that "physical bonding" is responsible. In my mind, there is no such thing as "physical bonding". To me, "bonding" is something permanent. The authors did not consider the role of nanowire-to-nanowire friction during deformation. Consider that millions of such nanowire-to-nanowire junctions preventing relative motion! Like in Velcro! In current form, the paper can be accepted, but there are two minor issues. First, the document should undergo thorough proofreading. I see issues of English grammar in quite a few places, especially in the new text inserted in response to reviewer comments. Second, the issue of reporting of data and the precision used to report mean and standard deviation. For example, if the mean is reported as 20.3, should the standard deviation be reported as 2.4? My recommendation is to round off the mean to 20 and report standard deviation as 2.4. Please check all these.

I believe readers would enjoy this contribution.

Reviewer #3 (Remarks to the Author):

The authors have addressed all my comments and I suggest its publication without further revision.

Point-by-Point Response to the Reviewers' Comments

Title: Strong yet Flexible Ceramic Aerogel

Manuscript ID: NCOMMS-22-48377B-Z

COMMENTS TO AUTHOR:

Reviewer #1 (Remarks to the Author):

Thank you for the authors' response to the comments. Based on the N₂ sorption results, it is evident that the surface area is quite low. The low N₂ sorption data confirms the low mesoporosity of the ceramic aerogel materials. Given that the N₂ molecule can exclusively detect the mesopore structure, the average pore sizes (11.8 and 14.3 nm) calculated using the BJH method may not encompass the entire range of pore sizes. To gain a comprehensive understanding of the overall pore volume and size of the samples, I suggest combining this information with bulk density (Reichenauer, G. (2011). Structural characterization of aerogels. Aerogels handbook, 449-498.):

$$1/\text{bulk density} = V_{\text{total}} = V_{\text{micro\&meso}} + 1/\text{skeletal density}$$

$$D_{\text{average}} = 4 * V_{\text{total}} / S_{\text{BET}}$$

It would be beneficial if the authors could integrate the N₂ sorption, mercury intrusion porosimetry, and the total pore calculation outlined above to offer insights into potential pore size distributions. This approach goes beyond merely listing two methods and discussing their results in isolation.

Response: We thank the reviewer for the constructive suggestion. By using the approach that you provided here, we can understand the whole picture of the pore size distribution in the laminated aerogel. The revised content is shown below.

Revised Supplementary Discussion.

We analyzed the pore size distribution in the laminated aerogel by integrating the N₂ sorption, mercury intrusion porosimetry, and the total pore calculation. N₂ sorption can provide the size information of the mesopores. Mercury intrusion porosimetry can reflect the size information of the micropores. The total pore calculation is a beneficial supplement to the above two methods and can reflect the whole picture of pore size distribution. The details are discussed below.

N₂ sorption measurement. Fig. 2j and k shows the N₂ sorption isotherms at 77 K and pore size distribution derived from Barrett–Joyner–Halenda (BJH) analysis, respectively. The calculated specific surface areas and average pore sizes of the laminated aerogel are 15.4 m²/g and 11.8 nm, respectively. The mesopore volume per gram, V_{meso}, is 0.03 cm³/g.

Mercury intrusion porosimetry. In the present case, the laminated aerogel is a highly compressible material. The deformation during the intrusion can influence the results: when the pressure is too high, the structure of the aerogel might be destroyed and some pores could be

compressed or disappear and cannot be detected by mercury intrusion porosimetry. From the compressive property of the laminated aerogel, we can see that the maximum stress can reach an average value of 1255 ± 116.3 kPa at 80% strain. It is sure that before the pressure of mercury reaches more than 1255 kPa the structure of the laminated aerogel is integral, except for the elastic deformation.

Herein, we assume that the pore shape is spherical, then the volume shrinkage ratio, X , during the mercury intrusion process can be calculated by equation 1.

$$X = \frac{\frac{4}{3}\pi R^3 - \frac{4}{3}\pi r^3}{\frac{4}{3}\pi R^3} \quad (1)$$

Where R is the real pore radius of the pores, r is the tested pore radius. The real pore radius be calculated by equation 2.

$$R = \left(\frac{1}{1-X}\right)^{\frac{1}{3}}r \quad (2)$$

Because of the nearly zero Poisson ratio, the volume shrinkage at $x\%$ compressive strain is about $x\%$. If there is no volume shrinkage, $R=r$. If the volume shrinkage is 60%, then $R=1.35r$; which indicates that the real pore size is 1.35 times that of the tested one. If the volume shrinkage is 80%, then then $R=1.71r$.

Supplementary Fig. 13 shows the relationship between the tested pore size and the applied pressure to the mercury (in the range of 0 to 1255 kPa). It could be seen that the maximum tested pore size is 355 μm (the pressure of mercury is 3.5 kPa) and the minimum tested pore size is 1 μm (the pressure of mercury is 1255 kPa). When the pressure of mercury exceeds 1255, the pore size might be compressed further. And with the further increase of the pressure, the structure might be destroyed. Therefore, the pore sizes measured here are smaller than the real ones and some pores with sizes in the smaller value range might not be tested.

Combining the mercury intrusion porosimetry results displayed in Supplementary Fig. 14, it could be concluded that the pore size in the laminated aerogel is below 355 μm . The micropore volume ratio measured by mercury intrusion porosimetry, $V_{\text{micro}-m}$, is $15.68 \text{ cm}^3/\text{g}$. Worth noting that the $V_{\text{micro}-m}$ measured here is smaller than the real one, V_{micro} , due to the volume shrinkage of the aerogel during the test. Therefore, there would be a “volume of missing”, V_{missing} . And V_{micro} can be written as equation 3.

$$V_{\text{micro}} = V_{\text{micro}-m} + V_{\text{missing}} \quad (3)$$

Supplementary Fig. 13 | The relationship between the tested pore size and the applied pressure to the mercury (in the range of 0 to 1250 kPa).

Supplementary Fig. 14 | The tested pore size distribution in the laminated aerogel by mercury intrusion porosimetry.

Total pore calculation and analysis. The bulk density of the present aerogel, ρ_{bulk} , is about 50 mg/cm³. The total pore volume per gram is calculated to be about 20 cm³/g by equation 4.

$$V_{total} = \frac{1}{\rho_{bulk}} \quad (4)$$

In an aerogel,

$$V_{total} = V_{meso} + V_{micro} + \frac{1}{\rho_{skeletal}} \quad (5)$$

where $\rho_{skeletal}$ is the density of the skeletal material. The skeletal material in the laminated aerogel is SiC-SiO_x nanowire. The density of SiO_x is about 2.8g/cm³, and the density of SiC is 3.2g/cm³.

Therefore, the density of the skeletal material can be estimated to be 3.0g/cm³. $\frac{1}{\rho_{skeletal}}$ is calculated to be 0.33 cm³/g.

In the present case,

$$V_{total} = V_{meso} + V_{micro-m} + V_{missing} + \frac{1}{\rho_{skeletal}} \quad (6)$$

then $V_{missing}$ is calculated to be 3.96 cm³/g, and V_{micro} is 19.64 cm³/g.

As a result, in the laminated aerogel, V_{total} is 20 cm³/g, V_{meso} is 0.03 cm³/g, V_{micro} is 19.64 cm³/g. The ratio of mesopore volume in the laminated aerogel is only 0.15%, showing the low mesoporosity in the laminated aerogel. According to Ref. 1, the average pore size of the aerogel calculated by equation 7 is 5.1 μm. These results show that the pores in the laminated aerogel are mainly micropores with sizes below 355 μm and the average pore size of all the pores is about 5.1 μm.

$$D_{average} = \frac{4 \times (V_{total} - \frac{1}{\rho_{skeletal}})}{S_{BET}} \quad (7)$$

Reference

1. Reichenauer G. Structural characterization of aerogels. *Aerogels handbook*, 449–498 (2011).

Revised content.

“Fig. 2j and k show the N₂ sorption isotherms at 77 K and pore size distribution derived from Barrett–Joyner–Halenda (BJH) analysis, respectively. The calculated specific surface areas of the laminated aerogel and raw aerogel are 15.4 and 29.1 m²/g, respectively. **The average sizes of the mesopores** are 11.8 and 14.3 nm, respectively. Both the specific surface area and pore size of the laminated aerogel show **decreases** when compared with those of the raw aerogel. The decrease **in** the specific surface area is related to the severe volume shrinkage of the raw aerogel during the preparation process, which results from the more compact interconnection between neighboring nanowires. **Given that the N₂ molecule can exclusively detect the mesoporous structure, the average pore sizes calculated using the BJH method may not encompass the entire range of pore sizes. To gain a comprehensive understanding of the overall pore volume and size of the samples, we then analyzed the pore distribution in the laminated aerogel by combining the N₂ sorption results with mercury intrusion porosimetry and total pore calculation³⁶. As discussed in the Supplementary Discussion, mesopore volume accounts for only 0.15% of the total pore volume, most of the pores are micropores with a size smaller than 355 μm, and the average size of all the pores is about 5.1 μm.”**

Added Ref.

36. Reichenauer G. Structural characterization of aerogels. *Aerogels handbook*, 449–498 (2011).

Reviewer #2 (Remarks to the Author):

I read this contribution with interest. I also went through reviewer comments and the author responses. I believe there is something unique in these materials. The method of aerogel fabrication by putting together thin nanowire aerogel paper is deceptively simple. This is where readers would ask why such materials show the properties that are reported in this paper. The authors indicated that "physical bonding" is responsible. In my mind, there is no such thing as "physical bonding". To me, "bonding" is something permanent. The authors did not consider the role of nanowire-to-nanowire friction during deformation. Consider that millions of such nanowire-to-nanowire junctions preventing relative motion! Like in Velcro!

In current form, the paper can be accepted, but there are two minor issues. First, the document should undergo thorough proofreading. I see issues of English grammar in quite a few places, especially in the new text inserted in response to reviewer comments. Second, the issue of reporting of data and the precision used to report mean and standard deviation. For example, if the mean is reported as 20.3, should the standard deviation be reported as 2.4? My recommendation is to round off the mean to 20 and report standard deviation as 2.4. Please check all these.

I believe readers would enjoy this contribution.

Response: Dear Reviewer, we thank you for the constructive and positive response to our revision. According to your suggestions,

1) we have proof-read the manuscript and corrected the grammar typos and the mean and standard deviation of the data. These changes were marked in red in the revised manuscript.

2) we also revised the description of the interlayer connection in the aerogel as well as minor revisions in the Deformation Mechanism part by adding the consideration of the nanowire-to-nanowire friction. The revision is shown below.

“We characterized the interlayer adhesion by testing a piece of the aerogel by applying tensile stress perpendicular to the nanowire layers. The result in Fig. 2i shows that the maximum adhesion stress is about 2.25 kPa. Even though the aerogel was delaminated during the test, some local parts between the neighboring layers were connected with each other. To insights into the microscale interconnection between neighboring layers, we observed the interface during the delamination process. As shown in Supplementary Fig. 2, during the delamination process, bridging behaviors of the nanowires were seen clearly at the interface between the neighboring laminates, showing a Velcro-like connection. Such kind of connection is formed during the sample preparation process. When two pieces of the paper-like raw nanowire aerogel connected with each other, the nanowires in one layer would be embedded in the pores in the other layer. During the evaporation of the solvent, the shrinkage of the raw nanowire aerogel took place. Under the action of the capillary force, the embedded nanowires were gradually trapped in the pores and fastened by the nanowires around the pores gradually. Therefore, after the evaporation of the solvents, a large number of nanowire-to-nanowire junctions formed at the interface. During the delamination, the nanowire-to-nanowire friction thus prevents the relative motion of the nanowires and nanowire layers, forming a good interconnection.”

Reviewer #3 (Remarks to the Author):

The authors have addressed all my comments and I suggest its publication without further revision.

Response: We thank the reviewer for the positive feedback.

Reviewer #1 (Remarks to the Author):

I appreciate the author's efforts to improve the paper according to the comments, I do not have further comments.

Title: Strong yet Flexible Ceramic Aerogel
Manuscript ID: NCOMMS-22-48377C

REVIEWERS' COMMENTS

Reviewer #1 (Remarks to the Author):

I appreciate the author's efforts to improve the paper according to the comments, I do not have further comments.

Response: Thank you very much for your suggestions in improving the quality of our work.